# Polymerization of C9 enhances bacterial cell envelope damage and killing by membrane attack complex pores

**Dennis J. Doorduijn**, **Dani A. C. Heesterbeek**, **Maartje Ruyken**, **Carla J. C. de Haas**, **Daphne A. C. Stapels**, **Piet C. Aerts**, **Suzan H. M. Rooijakkers**, **Bart W. Bardoel***

Department of Medical Microbiology, University Medical Center Utrecht, Utrecht, The Netherlands

* b.w.bardoel-2@umcutrecht.nl

**Data Availability Statement:** All relevant data are within the manuscript and its Supporting Information files.

## Abstract

Complement proteins can form membrane attack complex (MAC) pores that directly kill Gram-negative bacteria. MAC pores assemble by stepwise binding of C5b, C6, C7, C8 and finally C9, which can polymerize into a transmembrane ring of up to 18 C9 monomers. It is still unclear if the assembly of a polymeric-C9 ring is necessary to sufficiently damage the bacterial cell envelope to kill bacteria. In this paper, polymerization of C9 was prevented without affecting binding of C9 to C5b-8, by locking the first transmembrane helix domain of C9. Using this system, we show that polymerization of C9 strongly enhanced damage to both the bacterial outer and inner membrane, resulting in more rapid killing of several *Escherichia coli* and *Klebsiella* strains in serum. By comparing binding of wildtype and 'locked' C9 by flow cytometry, we also show that polymerization of C9 is impaired when the amount of available C9 per C5b-8 is limited. This suggests that an excess of C9 is required to efficiently form polymeric-C9. Finally, we show that polymerization of C9 was impaired on complement-resistant *E. coli* strains that survive killing by MAC pores. This suggests that these bacteria can specifically block polymerization of C9. All tested complement-resistant *E. coli* expressed LPS O-antigen (O-Ag), compared to only one out of four complement-sensitive *E. coli*. By restoring O-Ag expression in an O-Ag negative strain, we show that the O-Ag impairs polymerization of C9 and results in complement-resistance. Altogether, these insights are important to understand how MAC pores kill bacteria and how bacterial pathogens can resist MAC-dependent killing.

## Author summary

In this paper, we focus on how complement proteins, an essential part of the immune system, kill Gram-negative bacteria via so-called membrane attack complex (MAC) pores. The MAC is a large pore that consists of five different proteins. The final component, C9, assembles a ring of up to 18 C9 molecules that damages the bacterial cell envelope. Here, we aimed to better understand if this polymeric-C9 ring is necessary to kill bacteria and if bacteria can interfere in its assembly. We uncover that polymerization of C9 increased the

**Funding:** This work was funded by the European Research Council (ERC) Starting Grant 639209-ComBact (https://erc.europa.eu/funding/starting-grants (to SHMR), the Utrecht Molecular Immunology HUB (https://www.uu.nl/en/research/life-sciences/collaborate/hubs/utrecht-molecular-immunology-hub (to SHMR) and Aspasia grant (Dutch Research Council NWO, to SHMR). The funders had no role in study design, data collection and analysis, decision to publish, or preparation of the manuscript.

**Competing interests:** The authors have declared that no competing interests exist.

damage to the entire bacterial cell envelope, which resulted in more rapid killing of several Gram-negative species. We also show that some clinical *Escherichia coli* strains can block polymerization of C9 and survive MAC-dependent killing by modifying sugars in the bacterial cell envelope, namely the O-antigen of lipopolysaccharide. These insights help us to better understand how the immune system kills bacteria and how pathogenic bacteria can survive killing.

## Introduction

Complement proteins in human serum play a crucial role in fighting off invading bacteria. Activation of the complement cascade ultimately results in the assembly of membrane attack complex (MAC) pores that can directly kill Gram-negative bacteria [1–3]. MAC assembly is initiated when recognition molecules, such as antibodies and lectins, bind to bacteria and recruit early complement proteins [4]. This triggers a proteolytic cascade that deposits convertase enzymes on the bacterial surface [5]. These convertases convert complement component C5 into C5b, which initiates the assembly of a large ring-shaped MAC pore that damages the bacterial cell envelope [3,6,7]. Although MAC pores can efficiently kill complement-sensitive bacteria, some bacterial pathogens can survive killing by MAC pores [8–11]. Therefore, studying how MAC pores kill bacteria is important to understand how the complement system prevents infections and how bacterial pathogens resist killing by MAC pores.

MAC pores assemble in a stepwise manner [12]. When a surface-bound convertase converts C5 into C5b, C5b immediately binds to C6 to form the C5b6 complex [13,14]. Direct binding of C7 to C5b6 anchors the MAC precursor to the membrane [15]. Next, C8 binds to membrane-anchored C5b-7, which triggers structural rearrangements in C8 that result in insertion of a transmembrane β-hairpin into the membrane [16,17]. Finally, C9 binds to C5b-8 to form C5b-9$_1$, after which C9 self-polymerizes to form a transmembrane ring consisting of C5b-8 and up to 18 monomers of C9 with an inner diameter of 11 nm [18,19].

Although C5b-8 can already cause small 1–2 nm lesions in the membrane of erythrocytes and liposomes without polymeric-C9 [17,20], it is still unclear if C9 polymerization is required to sufficiently damage the complex bacterial cell envelope to kill Gram-negative bacteria. On bacteria, MAC pores assemble on the outer membrane (OM), which largely consists of lipopolysaccharide (LPS). The O-antigen (O-Ag) of LPS can vary in length between bacterial strains and species [21,22], and this has frequently been associated with complement-resistance [8,9,23,24]. Apart from the OM, the Gram-negative cell envelope also consists of a cytosolic inner membrane (IM) and a periplasmic peptidoglycan layer [25]. We recently developed methods to separately study OM and IM damage in time [14,26,27], which highlighted that C9 is required to efficiently trigger damage to the IM and to allow passage of antimicrobial enzymes in serum through the OM. Although the presence of polymeric-C9 has been associated with cell envelope damage [28], direct evidence to what extent polymerization of C9 contributes to bacterial cell envelope damage by MAC pores is limited. In addition, a direct causal link between bacterial killing and polymerization of C9 has still not been established, mainly because tools to specifically prevent polymerization of C9 have been limited [29]. Recently, Spicer *et al.* demonstrated that 'locking' the first transmembrane helix (TMH-1) domain of C9 could prevent polymerization of C9 [30]. Here, we used this 'locked' C9 to study how C9 polymerization contributes to bacterial cell envelope damage and killing by MAC pores.

In this paper, we show that polymerization of C9 enhanced the efficiency by which MAC pores damage both the OM and IM, which ultimately resulted in faster killing of several

*Escherichia coli* and *Klebsiella* strains. This study therefore highlights that polymerization of C9 is required to form MAC pores that efficiently damage the bacterial cell envelope and kill bacteria. Moreover, we found that polymerization of C9, but not binding of C9 to C5b-8, was impaired on several complement-resistant *E. coli* strains that survive killing by MAC pores. Finally, we show that the expression of LPS O-Ag impairs polymerization of C9 and results in resistance to MAC-dependent killing. This study therefore also provides insights into how bacterial pathogens resist MAC-dependent killing.

## Results

### Locking the TMH-1 domain of C9 strongly impairs its capacity to polymerize, without preventing binding to C5b-8 on *E. coli*

To study the contribution of C9 polymerization to bacterial killing by the MAC, we wanted to use a system in which C9 can bind to C5b-8, but cannot polymerize. Spicer *et al*. recently demonstrated that the TMH-1 domain of C9 has a crucial role in C9 polymerization [30]. When C9 binds to C5b-8, structural rearrangements in C9 trigger unfurling of the TMH-1 (**Fig 1A-I**) and TMH-2 to form a transmembrane β-hairpin. Although both TMH domains of C9 insert into the membrane, unfurling of the TMH-1 domain also exposes an elongation surface that allows a subsequent C9 to bind (**Fig 1A-II**). This ultimately results in the assembly of a polymeric-C9 ring that forms the MAC pore together with C5b-8 (**Fig 1A-III**). Based on this crucial role of the TMH-1 domain in polymerization, Spicer *et al*. designed a C9 TMH-1 'lock' mutant (C9$_{TMH-1 lock}$) in which the TMH-1 domain was linked to β-strand 4 of the MACPF/CDC domain via an intramolecular cysteine bridge. This lock prevents unfurling of the TMH-1 domain, and thus prevents both the formation of a transmembrane β-hairpin (**Fig 1A-IV**) and binding of a subsequent C9 (**Fig 1A-V**). Reducing the cysteine bridge with DTT can unlock the TMH-1 domain and restore its capacity to polymerize (**Fig 1A-VI**).

C9$_{TMH-1 lock}$ was recombinantly expressed and site-specifically labelled with a fluorophore via sortagging, as was done previously for wildtype C9 (C9$_{wt}$) [14]. Fluorescent labelling was comparable between both C9$_{TMH-1 lock}$ and C9$_{wt}$ (**S1A Fig**), which means that the fluorescence of both proteins can be directly compared in our assays. C9$_{TMH-1 lock}$ showed impaired lysis of sheep erythrocytes compared to C9$_{wt}$, which could be restored by reducing C9$_{TMH-1 lock}$ with DTT (**S1B Fig**). Moreover, fluorescent C9$_{TMH-1 lock}$ was used to distinguish SDS-stable polymeric-C9 from monomeric-C9 by SDS-PAGE, which is frequently used as a read-out for C9 polymerization [31]. C9$_{TMH-1 lock}$ did not form polymeric-C9 together with preassembled C5b6 (pC5b6), C7 and C8, whereas C9$_{wt}$ did (**S1C Fig**). These data confirm that the capacity of C9$_{TMH-1 lock}$ to form polymers is strongly impaired, although it has to be noted that due to the detection limit of polymeric-C9 by SDS-PAGE we cannot state that polymerization of C9 is completely prevented.

Although Spicer *et al*. already demonstrated that locking the TMH-1 domain prevents polymerization of C9 [30], we wanted to confirm that this did not prevent binding of C9 to C5b-8. *E. coli* MG1655 bacteria were incubated in C8-depleted serum to activate complement and label them with MAC precursor C5b-7 (**Fig 1B**). C5b-7 labelled bacteria were washed to remove remaining serum components and incubated with C8 and C9$_{wt}$ or C9$_{TMH-1 lock}$ to further assemble the MAC. Both C9$_{wt}$ and C9$_{TMH-1 lock}$ bound to C5b-7 in a C8-dependent manner as measured by flow cytometry (**Fig 1C**), but C9$_{wt}$ binding was 10-fold higher than C9$_{TMH-1 lock}$. Since the amount of C5b-8 on the surface, as measured by C6-FITC binding, was comparable for both C9$_{wt}$ and C9$_{TMH-1 lock}$ (**S1D Fig**), the relative difference in C9 binding suggested a difference in polymerization of C9 (**Fig 1D**). SDS-PAGE confirmed that only monomeric-C9 was detected on bacteria incubated with C9$_{TMH-1 lock}$ (**Fig 1E**). Reducing

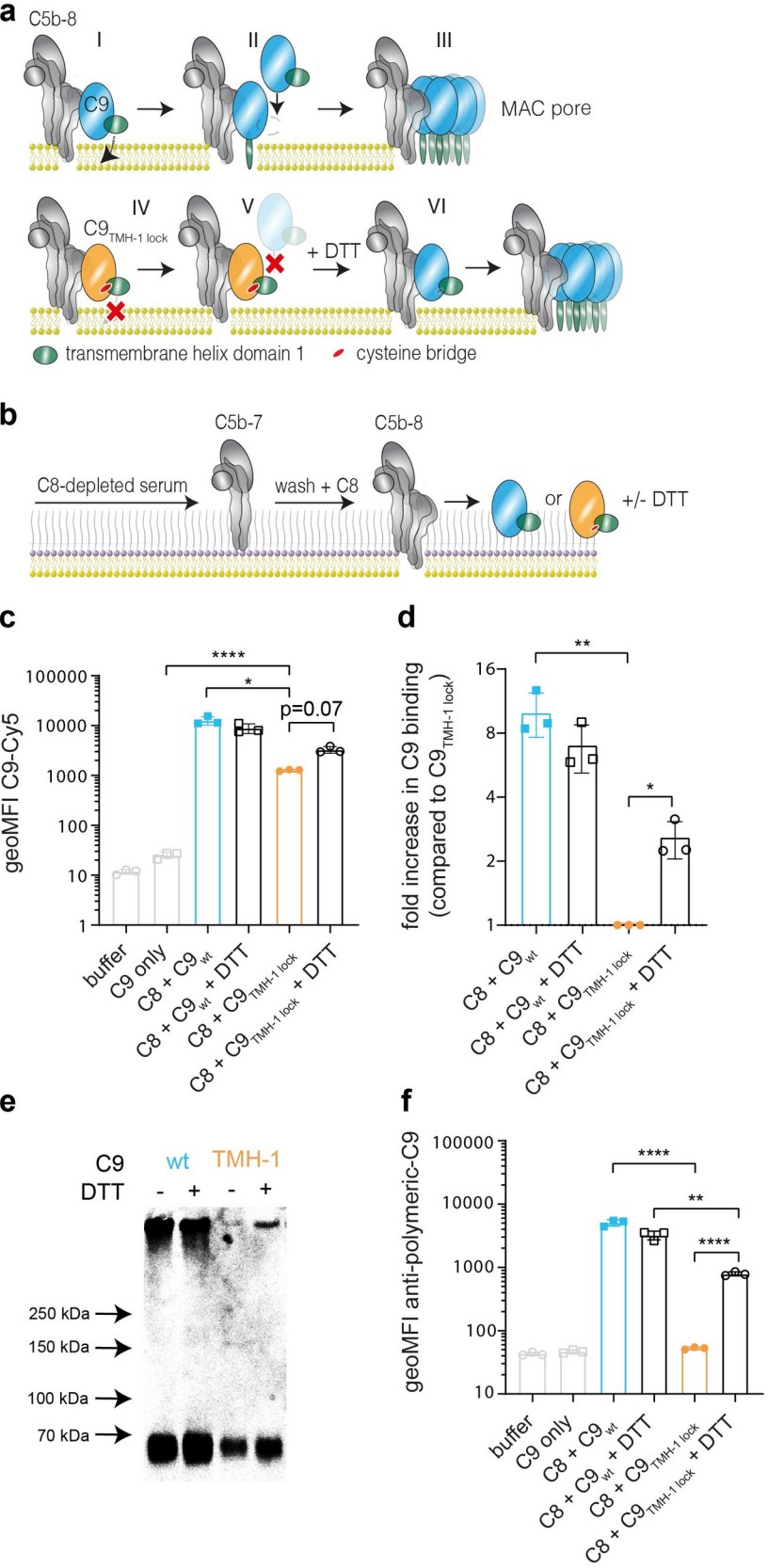

**Fig 1. C9$_{TMH-1\ lock}$ binds to C5b-8, but its capacity to form polymers is impaired on *E. coli*.** a) Schematic overview how C9 polymerizes. Binding of C9 (blue) to C5b-8 (grey) triggers unfurling of the TMH-1 domain of C9 (green) and subsequent insertion into the membrane (I). Unfurling of TMH-1 exposes the elongation surface of C9 to bind a subsequent C9 monomer (II). Ultimately, this results in the formation of a C9 polymer (III). Locking the TMH-1 domain of C9 (orange, C9$_{TMH-1\ lock}$) with an intramolecular cysteine bridge (red) prevents unfurling of the TMH-1 domain (IV). This also prevents exposure of the elongation surface of C9$_{TMH-1\ lock}$ and subsequent polymerization (V). Reducing the cysteine bridge of C9$_{TMH-1\ lock}$ with DTT restores its polymerizing capacity (VI). b) Schematic overview how *E. coli* MG1655 were labelled with MAC. Bacteria were labelled with C5b-7 by incubating them in 10% C8-depleted serum for 30 minutes. Bacteria are washed and next incubated with 10 nM C8 for 15 minutes. Finally, 20 nM of Cy5-labelled C9$_{wt}$ or C9$_{TMH-1\ lock}$ is added in the presence or absence of 10 mM DTT for 30 minutes. c) Binding of Cy5-labelled C9$_{wt}$ or C9$_{TMH-1\ lock}$ to bacteria treated as described in b and measured by flow cytometry. d) Binding of Cy5-labelled C9 in c was divided by the fluorescence of bacteria labelled with C8 + C9$_{TMH-1\ lock}$ to calculate the relative binding difference as indication for polymerization of C9. e) Bacterial cell pellets were analyzed by SDS-PAGE for in-gel fluorescence of Cy5-labelled C9$_{wt}$ or C9$_{TMH-1\ lock}$ to distinguish monomeric-C9 from polymeric-C9. f) Bacteria were washed after DTT incubation and next stained with AF488-labelled mouse anti-polymeric-C9 antibody (aE11) and staining was measured by flow cytometry. Flow cytometry data are represented by geoMFI values of the bacterial population. The SDS-PAGE image is representative for at least three independent experiments. Data represent individual values of three independent experiments with mean +/- SD. Statistical analysis was done on log-transformed data ($^{10}$log for c and f, $^{2}$log for d) using a paired one-way ANOVA with Tukey's multiple comparisons' test. Significance was shown as * $p \leq 0.05$, ** $p \leq 0.005$, **** $p \leq 0.0001$.

C9$_{TMH-1\ lock}$ with DTT increased C9 binding 3-fold compared to C9$_{TMH-1\ lock}$ without DTT (**Fig 1C** and **1D**) and resulted in the detection of polymeric-C9 on bacteria by SDS-PAGE (**Fig 1E**). Both C9 binding and polymeric-C9 detection were approximately 2- to 3-fold lower compared to C9$_{wt}$ with DTT (**Fig 1D** and **1E**), suggesting that reducing C9$_{TMH-1\ lock}$ only partially restored its capacity to polymerize compared to C9$_{wt}$. Finally, an antibody that recognizes a neo-epitope exposed in polymeric-C9, which is frequently used for the detection of MAC pores [32], specifically detected bacteria incubated with C9$_{wt}$ and C9$_{TMH-1\ lock}$ with DTT, but not with C9$_{TMH-1\ lock}$ without DTT (**Fig 1F**). In summary, these data indicate that C9$_{TMH-1\ lock}$ can bind to C5b-8 on *E. coli*, but that its capacity to polymerize is strongly impaired.

## Polymerization of C9 enhances bacterial killing by MAC pores

We then assessed if polymerization of C9 is important for bacterial killing by MAC pores. A DNA dye (Sytox) that cannot permeate an intact IM was used to measure the percentage of cells with IM damage by flow cytometry, which we have previously shown to be a sensitive read-out for bacterial killing [14]. Adding C9$_{wt}$ to C5b-8 labelled bacteria resulted in IM damage in a dose-dependent manner, reaching 100% Sytox positive cells from above 3 nM C9 (**Fig 2A**). For C9$_{TMH-1\ lock}$, IM damage was impaired and did not increase above 30% Sytox positive cells at 100 nM C9 (**Fig 2A**). Moreover, bacterial viability was determined by counting colony forming units (CFU's) and decreased only 10-fold for C9$_{TMH-1\ lock}$ compared to C5b-8 alone, whereas C9$_{wt}$ decreased bacterial viability at least a 1,000-fold (**Fig 2B**). Reducing C9$_{TMH-1\ lock}$ with DTT restored its capacity to damage the IM (**Fig 2A**) and kill bacteria (**Fig 2B**) relative to C9$_{wt}$ in reducing conditions. However, reducing C9$_{wt}$ or C9$_{TMH-1\ lock}$ did diminish IM damage (**Fig 2A**) and bacterial killing (**Fig 2B**) compared to C9$_{wt}$ in non-reducing conditions. Unlocking the TMH-1 domain after C9$_{TMH-1\ lock}$ has bound C5b-8, and the remaining soluble C9$_{TMH-1\ lock}$ was washed away, did not further increase IM damage (**S2A Fig**).

Polymerization of C9$_{wt}$ (**Fig 2C**) and subsequent IM damage (**Fig 2D**) could be inhibited by C9$_{TMH-1\ lock}$ in a dose-dependent manner. Flow cytometry revealed that polymeric-C9 already decreased by 50% when the amount of C9$_{TMH-1\ lock}$ was still 10-fold lower than the amount of C9$_{wt}$ (**Fig 2C**). This suggests that C9$_{TMH-1\ lock}$ can interfere at multiple stages in the assembly of a polymeric-C9 ring. However, IM damage was only fully inhibited when there was 10-fold more C9$_{TMH-1\ lock}$ than C9$_{wt}$ (**Fig 2D**). This indicates that little polymeric-C9 is required to damage the IM. This raised the question whether few ring-shaped MAC pores with 18 C9

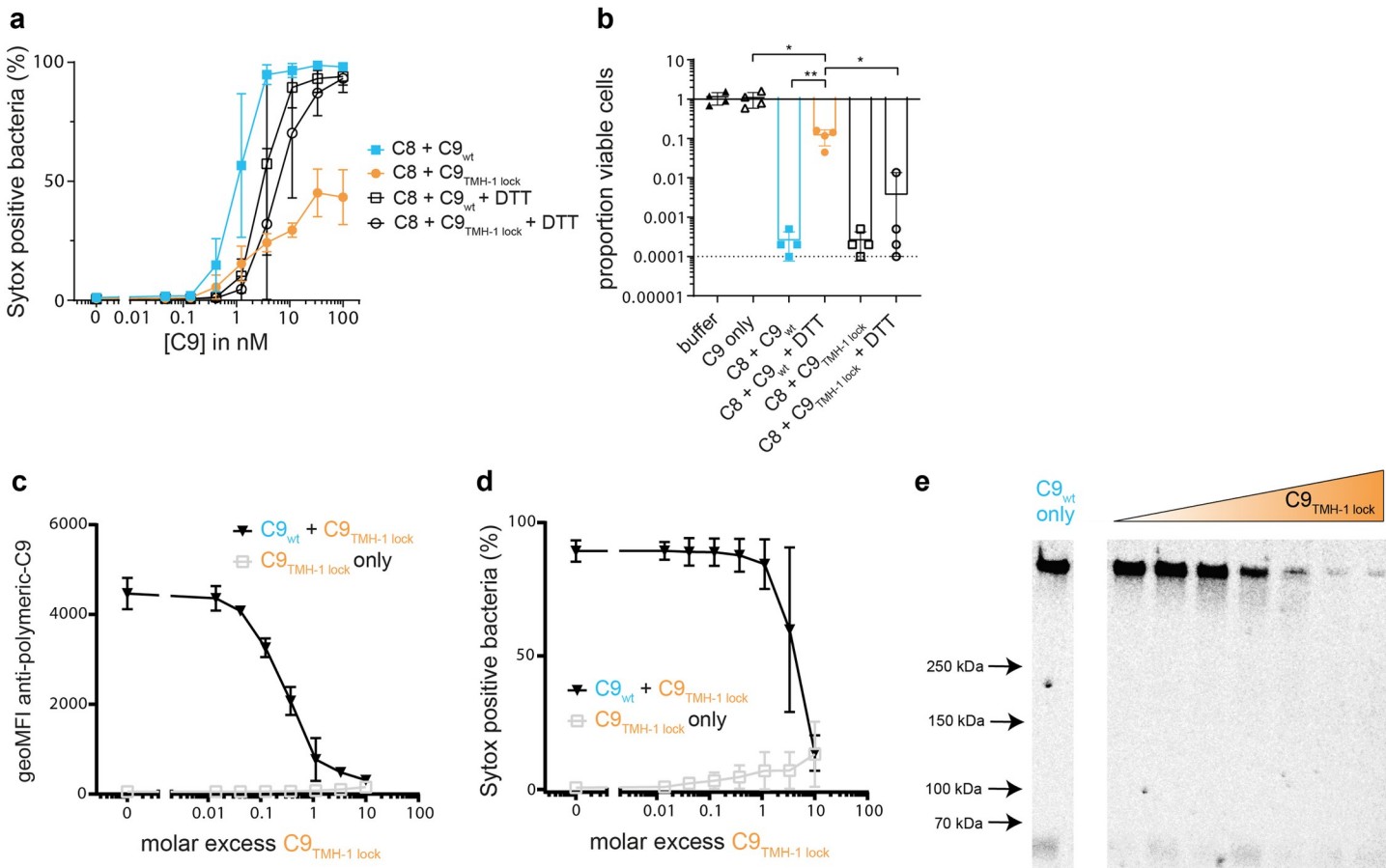

**Fig 2. Polymerization of C9 enhances bacterial killing by MAC pores.** *E. coli* MG1655 was labelled with C5b-7 by incubating in 10% C8-depleted serum for 30 minutes. Bacteria were washed and next incubated with 10 nM C8 for 15 minutes. a-b) A concentration range of $C9_{wt}$ or $C9_{TMH-1\ lock}$ was added in the absence or presence of 10 mM DTT for 30 minutes. a) Sytox was used to determine the percentage of cells that have a damaged bacterial IM by flow cytometry as read-out for bacterial killing. b) Bacterial viability was determined by counting colony forming units (CFU's) and calculating the proportion of viable cells for 100 nM C9 condition (a) compared to C5b-7 labelled bacteria in buffer. The horizontal dotted line represents the detection limit of the assay. c-d) C5b-8 labelled bacteria were incubated for 30 minutes with 20 nM $C9_{wt}$ and a concentration range of $C9_{TMH-1\ lock}$. Bacteria were stained with AF488-labelled mouse anti-polymeric-C9 aE11-antibody (c) and Sytox to determine the percentage of cells that has a damaged bacterial IM (d) by flow cytometry. e) Bacterial cell pellets were analyzed by SDS-PAGE for in-gel fluorescence of Cy5-labelled $C9_{wt}$ to distinguish monomeric-C9 from polymeric-C9. From left to right, the molar excess of $C9_{TMH-1\ lock}$ from left to right is: 0 ($C9_{wt}$ only), 0.016, 0.05, 0.14, 0.46, 1.1, 3.3 and 10. The SDS-PAGE image is representative for at least three independent experiments. Flow cytometry data are represented by geoMFI values or cell frequencies of the bacterial population. Data represent mean values +/- SD (a, c and d) of three independent experiments or individual values (c) of four independent experiments with mean +/- SD. Statistical analysis was done on $^{10}$log-transformed data using a paired one-way ANOVA with Tukey's multiple comparisons' test (b). Significance was shown as $^*$ p $\leq$ 0.05, $^{**}$ p $\leq$ 0.005.

monomers, or arc-shaped MAC pores with fewer C9 monomers, as observed for other homologous pore forming proteins [33,34], can already damage the IM. SDS-PAGE revealed a similar decrease in polymeric-C9 (**Fig 2E**) as with staining by flow cytometry (**Fig 2C**). No incomplete C9 polymers were detected by SDS-PAGE, although it is unclear if these potential incomplete C9 polymers are SDS-stable. Altogether our data primarily suggest that polymerization of C9 enhances bacterial killing by MAC pores.

## Polymerization of C9 increases OM damage

Since MAC pores assemble on the OM [14], we also wondered how C9 polymerization contributes to OM damage. First, OM damage was compared for $C9_{wt}$ and $C9_{TMH-1\ lock}$ by

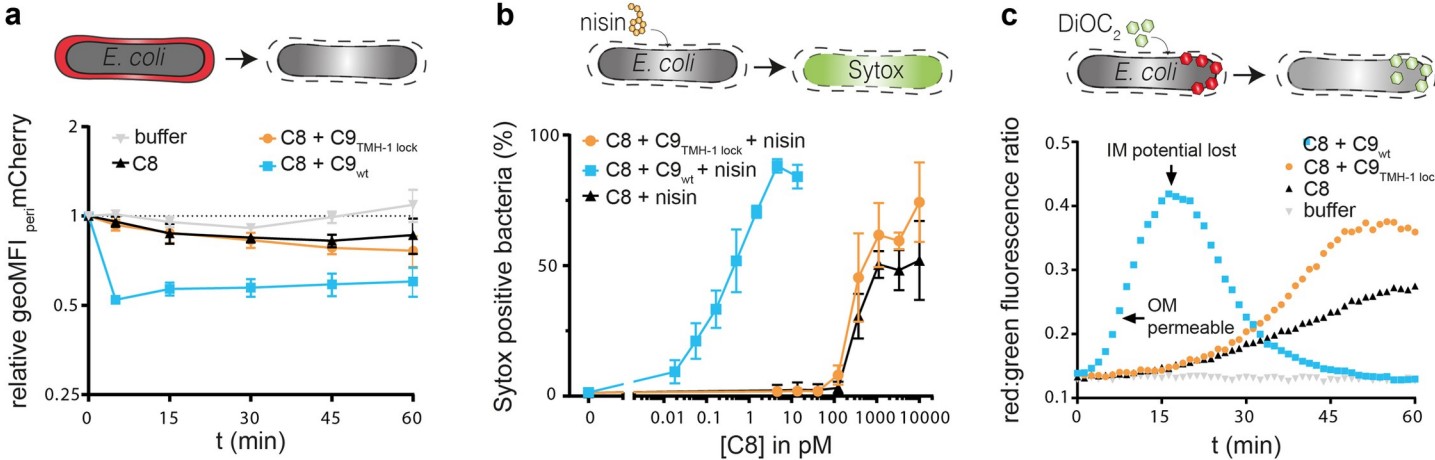

**Fig 3. Polymerization of C9 increases OM damage by MAC pores.** *E. coli* MG1655 (grey) was labelled with C5b-7 by incubating in 10% C8-depleted serum for 30 minutes. Bacteria were washed and next incubated with buffer, 10 nM C8 with 20 nM C9$_{wt}$ or C9$_{TMH-1 lock}$ to measure damage to the bacterial OM. a) periplasmic mCherry ($_{peri}$mCherry, red) leakage was measured at different time points by flow cytometry and represented as relative $_{peri}$mCherry fluorescence compared to t = 0. b) C5b-7 labelled bacteria were incubated with a concentration range of C8 and 20 nM C9$_{wt}$ or C9$_{TMH-1 lock}$ supplemented with 3 µg/ml nisin for 30 minutes. Nisin influx through OM was measured using Sytox to determine the percentage of cells that have a damaged bacterial IM as read-out for bacterial killing by flow cytometry after 30 minutes. c) DiOC$_2$ enters the periplasm when the OM is permeable and shifts from green to red fluorescence in cells with an intact inner membrane (IM) potential. DiOC$_2$ shifts back to green fluorescence when the IM potential is lost. Flow cytometry data are represented by geoMFI values of the bacterial population. Data represent mean +/- SD of three independent experiments (a and b). The multiwell plate-reader assay (c) is shown by one representative experiment that has been repeated at least three times.

measuring leakage of periplasmic mCherry ($_{peri}$mCherry, 22 kDa) through the OM of MG1655 using flow cytometry [14]. Adding C9$_{wt}$ to C5b-8 labelled bacteria resulted in rapid leakage of $_{peri}$mCherry within 5 minutes (**Fig 3A**). By contrast, no more $_{peri}$mCherry leaked out with C9$_{TMH-1 lock}$ compared to C5b-8 alone within 60 minutes (**Fig 3A**), or when the TMH-1 domain was unlocked after C9$_{TMH-1 lock}$ had bound C5b-8 and the remaining C9$_{TMH-1 lock}$ was washed away (**S2B Fig**). This suggests that polymerization of C9 is required to cause leakage of periplasmic proteins through the OM.

We also assessed if C9 polymerization affects the influx of extracellular molecules through the OM. We have previously shown that perturbation of the OM by MAC pores can sensitize Gram-negative bacteria to the antibiotic nisin (3.4 kDa), which normally cannot pass through the OM of Gram-negative bacteria [26]. A 1,000-fold more C5b-8 was required to sensitize bacteria to nisin with C5b-8 alone or C5b-8 + C9$_{TMH-1 lock}$ compared to C5b-8 + C9$_{wt}$ (**Fig 3B**), suggesting that polymerization of C9 strongly enhanced damage to the OM. Finally, we measured the influx of DiOC$_2$ (0.5 kDa) through the OM in the presence or absence of polymeric-C9. DiOC$_2$ is a green fluorescent dye that shifts to red fluorescence when it is incorporated in membranes with a membrane potential, which in the case of *E. coli* is the IM. With C9$_{TMH-1 lock}$, the increase in red:green fluorescence ratio was delayed compared to C9$_{wt}$, suggesting that polymerization of C9 enhanced the influx of DiOC$_2$ through the OM (**Fig 3C**). When bacteria die, the IM potential is lost, which results in a drop of red fluorescence. Loss of IM potential, initiated at the peak in red:green ratio, was also delayed with C9$_{TMH-1 lock}$ compared to C9$_{wt}$ (**Fig 3C**). Interestingly, C9$_{TMH-1 lock}$ did cause more rapid influx of DiOC$_2$ (**Fig 3C**) and nisin (**S3 Fig**) through the OM compared to C5b-8 alone. This suggests that C9$_{TMH-1 lock}$ can slightly increase OM damage. Nonetheless, together these data highlight that polymerization of C9 increases damage to the OM.

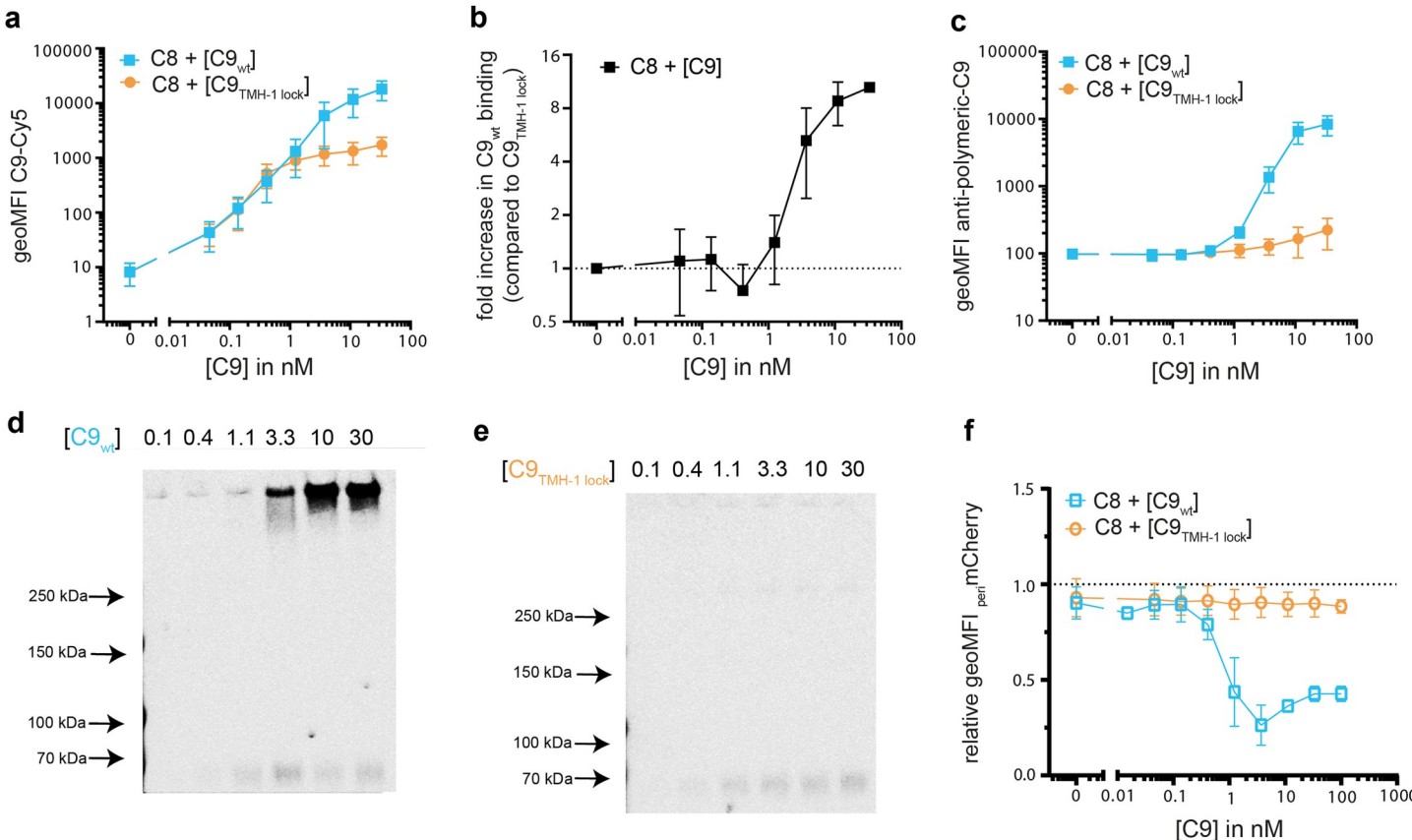

**Fig 4. An excess of C9 is required for efficient polymerization of C9 on bacteria.** *E. coli* MG1655 was labelled with C5b-7 by incubating in 10% C8-depleted serum for 30 minutes. Bacteria were washed and next incubated with 10 nM C8 and a concentration range of Cy5-labelled C9wt or C9TMH-1 lock for 30 minutes to measure binding of C9. a) Binding of Cy5-labelled C9wt or C9TMH-1 lock to bacteria measured by flow cytometry. b) The relative increase in C9wt binding compared to bacteria labelled with C9TMH-1 lock was calculated as indication for C9 polymerization. c) Bacteria were stained with AF488-labelled mouse anti-polymeric-C9 aE11-antibody and staining was measured by flow cytometry. Bacterial cell pellets were analyzed by SDS-PAGE for in-gel fluorescence of Cy5-labelled C9wt (d) or Cy5-labelled C9TMH-1 lock (e) to distinguish monomeric-C9 from polymeric-C9. f) periplasmic mCherry (perimCherry) leakage was measured by flow cytometry and represented as relative perimCherry fluorescence compared to t = 0. Flow cytometry data are represented by geoMFI values of the bacterial population. Data represent mean +/- SD of three independent experiments (a, b, c and f). SDS-PAGE images are representative for at least three independent experiments.

## An excess of C9 is required for efficient polymerization of C9 on bacteria

We next used C9TMH-1 lock to study the efficiency of C9 polymerization on bacteria when the amount of available C9 was limited. *E. coli* MG1655 were labelled with a constant amount of C5b-8 while the amount of available C9wt and C9TMH-1 lock was limited. Binding of C9wt was comparable to C9TMH-1 lock when the amount of available C9 was below 1 nM (**Fig 4A and 4B**). In addition, below 1 nM C9 no polymeric-C9 was detected via antibody staining (**Fig 4C**) and SDS-PAGE (**Fig 4D and 4E**) for both C9wt and C9TMH-1 lock. This suggests that there is little polymerization when the amount of available C9 is limited. As the amount of available C9 was increased above 1 nM C9, binding of C9wt continued to increase up to 10-fold higher than C9TMH-1 lock (**Fig 4A and 4B**), whereas binding of C9TMH-1 lock saturated (**Fig 4A**). This coincided with the detection of polymeric-C9 via antibody staining (**Fig 4C**) and SDS-PAGE (**Fig 4D**) for C9wt, whereas only monomeric-C9 was detected for C9TMH-1 lock (**Fig 4E**). In addition, perimCherry only leaked through the OM when the concentration of C9wt was above 1 nM (**Fig 4F**). Since leakage of perimCherry is dependent on the presence of polymeric-C9 (**Fig 3A**), these data suggest that C9 does not efficiently polymerize when the amount of available C9 is

limited. Finally, when the amount of surface-bound C5b-8 was varied by titrating C8, the difference in $C9_{wt}$ binding compared to $C9_{TMH-1\ lock}$ binding decreased when an excess of C8 was added (**S4A and S4B Fig**). $_{peri}$mCherry leakage (**S4C Fig**) and SDS-PAGE (**S4D Fig**) confirmed that an excess of C8 decreased the relative abundance of polymeric-C9 compared to monomeric-C9. This suggests that an increase in the amount of C5b-8 on the bacterial surface impairs polymerization of C9. In conclusion, these data suggest that an excess of C9 is required for efficient polymerization of C9 on bacteria.

## Polymerization of C9 enhances bacterial cell envelope damage and killing in serum

So far, we have looked at the effect of C9 polymerization on the MAC in the absence of other serum components. Next, we wanted to see if polymerization of C9 also enhances bacterial cell envelope damage and killing in a serum environment. *E. coli* MG1655 was incubated in 3% C9-depleted serum with $C9_{wt}$ or $C9_{TMH-1\ lock}$ and influx of $DiOC_2$ or Sytox were measured over time in a plate-reader to measure OM and IM damage respectively. OM damage (**Fig 5A**) preceded IM damage (**Fig 5B**), but both were delayed by 30 minutes with $C9_{TMH-1\ lock}$ compared to $C9_{wt}$. Also, bacterial viability was a 1,000-fold higher for $C9_{TMH-1\ lock}$ compared to $C9_{wt}$ after 30 minutes (**Fig 5C**). These data suggest that polymerization of C9 enhances bacterial cell envelope damage and killing in serum.

Interestingly, bacteria were killed with $C9_{TMH-1\ lock}$ to a comparable level as $C9_{wt}$ after 90 minutes. This was not observed with $C9_{TMH-1\ lock}$ in the absence of serum (**S5A Fig**), which indicated that the presence of serum affects MAC-dependent killing, even in the absence of polymerized C9. In serum, complement activation is still ongoing, which could ultimately result in more MAC pores on the surface of bacteria and explain why viability still decreases in time. Indeed, stopping further MAC formation, by adding C5 conversion inhibitor OmCI after 30 minutes, prevented bacterial killing in C9-depleted serum with $C9_{TMH-1\ lock}$ over a 100-fold (**Fig 5C**). This suggests that ongoing conversion of C5 and subsequent MAC formation is primarily responsible for the increase in bacterial killing over time. Serum also contains bactericidal enzymes that can more efficiently pass through a damaged OM, such as lysozyme (14.7 kDa) and type IIa secreted phospholipase 2A (PLA, 14.5 kDa) [27]. Both lysozyme and PLA decreased bacterial viability when C5b-8 labelled bacteria were incubated with $C9_{TMH-1\ lock}$ compared to C5b-8 alone (**S5B Fig**). This suggests that these serum proteins also contribute to MAC-dependent killing in serum in the absence of polymerized C9. Nonetheless, taken together our data mainly highlight that polymerization of C9 enhances the efficiency by which MAC pores damage the bacterial cell envelope and kill bacteria in serum.

## Polymerization of C9 enhances cell envelope damage for several *E. coli* and *Klebsiella* strains in serum

We wondered if our findings on *E. coli* MG1655 could also be extrapolated to other *E. coli* strains and other Gram-negative species. First, the effect of C9 polymerization on OM and IM damage was measured for three other complement-sensitive *E. coli* strains (BW25113, MC1061 and 547563.1) in serum. To compare strains, time points were interpolated at which the red:green fluorescence ratio of $DiOC_2$ reached half the value of the peak ($t_{1/2peak}$, shown in **Fig 5A**) as measure for OM damage, and at which half the maximum Sytox value was reached ($t_{1/2maximum}$, shown in **Fig 5B**) as measure for IM damage. Both OM damage (**Figs 5D** and **S6A**) and IM damage (**Figs 5E** and **S6B**) were delayed with $C9_{TMH-1\ lock}$ compared to $C9_{wt}$ for all tested *E. coli* strains (MG1655, BW25113, 547563.1 and MC1061). $C9_{TMH-1\ lock}$ did increase OM damage and IM damage compared to C9-depleted serum alone for three out of four *E.*

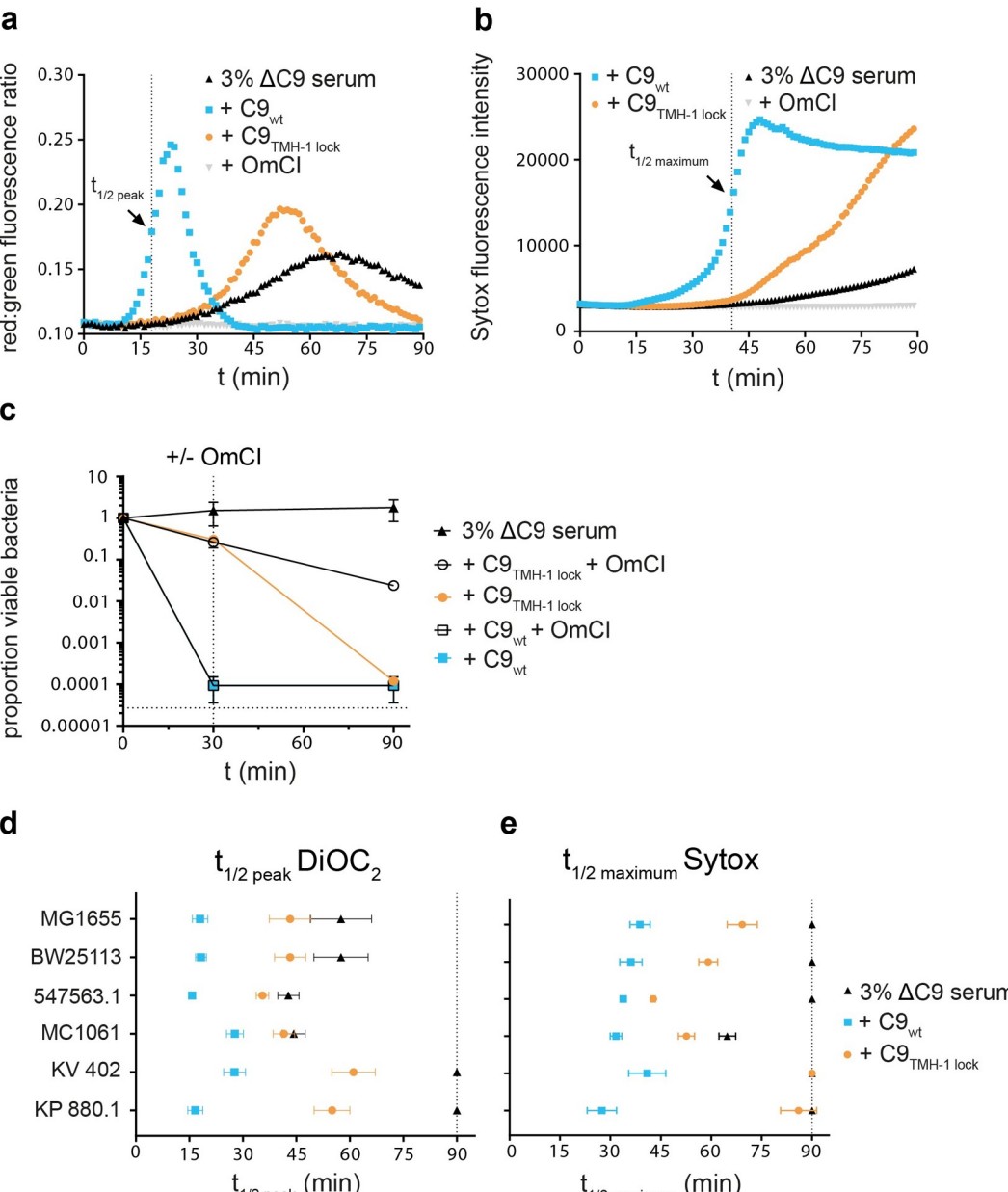

**Fig 5. Polymerization of C9 enhances bacterial cell envelope damage and killing in serum.** *E. coli* MG1655 was incubated in 3% C9-depleted serum supplemented with a physiological concentration (= 25 nM) of $C9_{wt}$ or $C9_{TMH-1\ lock}$ for 90 minutes. As negative control, 25 µg/ml C5 conversion inhibitor OmCI was added to the serum. a) OM damage was measured by $DiOC_2$ influx, which was determined by the shift in red:green fluorescence ratio over time in a multiwell plate-reader assay. b) IM damage was measured by Sytox influx over time in a multiwell plate-reader assay. c) Bacterial viability was determined at different time points by counting colony forming units (CFU's) and calculating the proportion of viable cells compared to t = 0. At t = 30 (vertical dotted line), buffer was added (closed symbols) or OmCI (open symbols) to stop MAC formation. The horizontal dotted line represents the detection limit of the assay. d-e) Other *E. coli* strains (BW25113, MC1061, 547563.1) and *K. variicola* 402 were also incubated in 3% C9-depleted serum supplemented with $C9_{wt}$ or $C9_{TMH-1\ lock}$ for 90 minutes. *K. pneumoniae* 567880.1 was incubated in 10% C9-depleted serum because of less efficient activation of the complement cascade and was therefore also supplemented with 80 nM $C9_{wt}$ or $C9_{TMH-1\ lock}$. d) OM damage was represented for all strains by the time when $DiOC_2$ influx reached half the value of the peak ($t_{1/2peak}$, shown in a for $C9_{wt}$). e) IM damage was represented for all strains by the time when Sytox fluorescence reached half the maximum value ($t_{1/2maximum}$, shown in b for $C9_{wt}$). Multiwell plate-reader assays (a and b) are shown by one representative experiment that has been repeated at least three times. Data represent mean +/- SD of three independent experiments (c, d and e).

*coli* strains (**Figs 5D and 5E**, **S6A and S6B**). For one strain, 547563.1, C9$_{TMH-1 lock}$ only slightly enhanced IM damage, since C9-depleted serum alone already damaged the IM of these bacteria (**Figs 5E and S6B**). OM damage (**Figs 5D and S6A**) and IM damage (**Figs 5E and S6B**) were also delayed with C9$_{TMH-1 lock}$ compared to C9$_{wt}$ for two clinical *Klebsiella* isolates (*Klebsiella variicola* 402 and *Klebsiella pneumoniae* 567880.1). C9$_{TMH-1 lock}$ did not cause IM damage within 90 minutes for *Klebsiella*, suggesting that IM damage by MAC pores for *Klebsiella* was more dependent on polymerization of C9 than for *E. coli*. In conclusion, these data suggest that C9 polymerization enhances cell envelope damage for multiple complement-sensitive *E. coli* and *Klebsiella* strains in serum.

## Polymerization of C9 is impaired on complement-resistant *E. coli* that express LPS O-Ag

In our previous study we have identified *E. coli* isolates that survive killing by MAC pores because MAC does not stably insert into the OM [15]. Here, we wanted to see if C9 polymerizes on these complement-resistant *E. coli* strains (clinical isolates 552059.1, 552060.1 and 567705.1). Binding of C9$_{TMH-1 lock}$ in 3% C9-depleted serum was comparable between all strains, suggesting that the total amount of C5b-8 on the surface was comparable for complement-sensitive and complement-resistant *E. coli* (**Fig 6A**). However, for complement-resistant *E. coli* binding of C9$_{wt}$ was similar to C9$_{TMH-1 lock}$, indicating that C9 did not polymerize (**Fig 6B**). By contrast, binding of C9$_{wt}$ was 5- to 15-fold higher than C9$_{TMH-1 lock}$ for all four complement-sensitive *E. coli* (**Fig 6B**). Adding up to 10-fold more C9 slightly increased the difference between C9$_{wt}$ binding and C9$_{TMH-1 lock}$ 2- to 3-fold on complement-resistant 552059.1 (**Fig 6C**), but this difference was still 5-fold lower compared to complement-sensitive MG1655. Although this indicated that some C9 polymerized on 552059.1, bacteria were still not killed (**S7A Fig**). These data suggest that polymerization of C9, but not binding of C9 to C5b-8, is impaired on complement-resistant *E. coli*.

We wondered if the LPS O-Ag is responsible for impairing polymerization of C9. The LPS O-Ag has frequently been associated with complement-resistance [23], although it remains unclear how. All three complement-resistant strains express LPS O-Ag (**S7B Fig**), whereas only one out of four complement-sensitive strains used in this study expresses LPS O-Ag. This indicated that the LPS O-Ag could be involved in impairing polymerization of C9. To directly study if O-Ag impairs polymerization of C9, we restored O-Ag expression in a complement-sensitive *E. coli* K12 strain (CGSC7740) that does not express O-Ag [35]. This strain lacks O-Ag because of an IS5-element that inactivates the *wbbL* gene that encodes an essential rhamnose transferase required for the expression of O-Ag [36]. Removal of this IS5-element restored O-Ag expression in CGSC7740 *wbbL+* (**S7C Fig**) and resulted in resistance to MAC-dependent killing in 5% serum (**S7D Fig**). Binding of C9$_{TMH-1 lock}$ in C9-depleted serum was comparable between CGSC7740 *wbbL+* and the wildtype strain (**Fig 6D**), suggesting that complement was activated in a comparable manner. However, binding of C9$_{wt}$ was only 2.5-fold higher than C9$_{TMH-1 lock}$ on CGSC7740 *wbbL+*, whereas C9$_{wt}$ bound 12-fold more compared to C9$_{TMH-1 lock}$ on the wildtype strain (**Fig 6E**). This suggests that polymerization of C9 is impaired on CGSC7740 *wbbL+*. In summary, these data highlight that expression of O-Ag results in resistance to MAC-dependent killing and impairs polymerization of C9.

## Discussion

Understanding how MAC pores damage the bacterial cell envelope and kill bacteria is important to understand how the complement system prevents infections. Although it was still unclear whether a completely assembled MAC pore is needed to kill bacteria, we here show

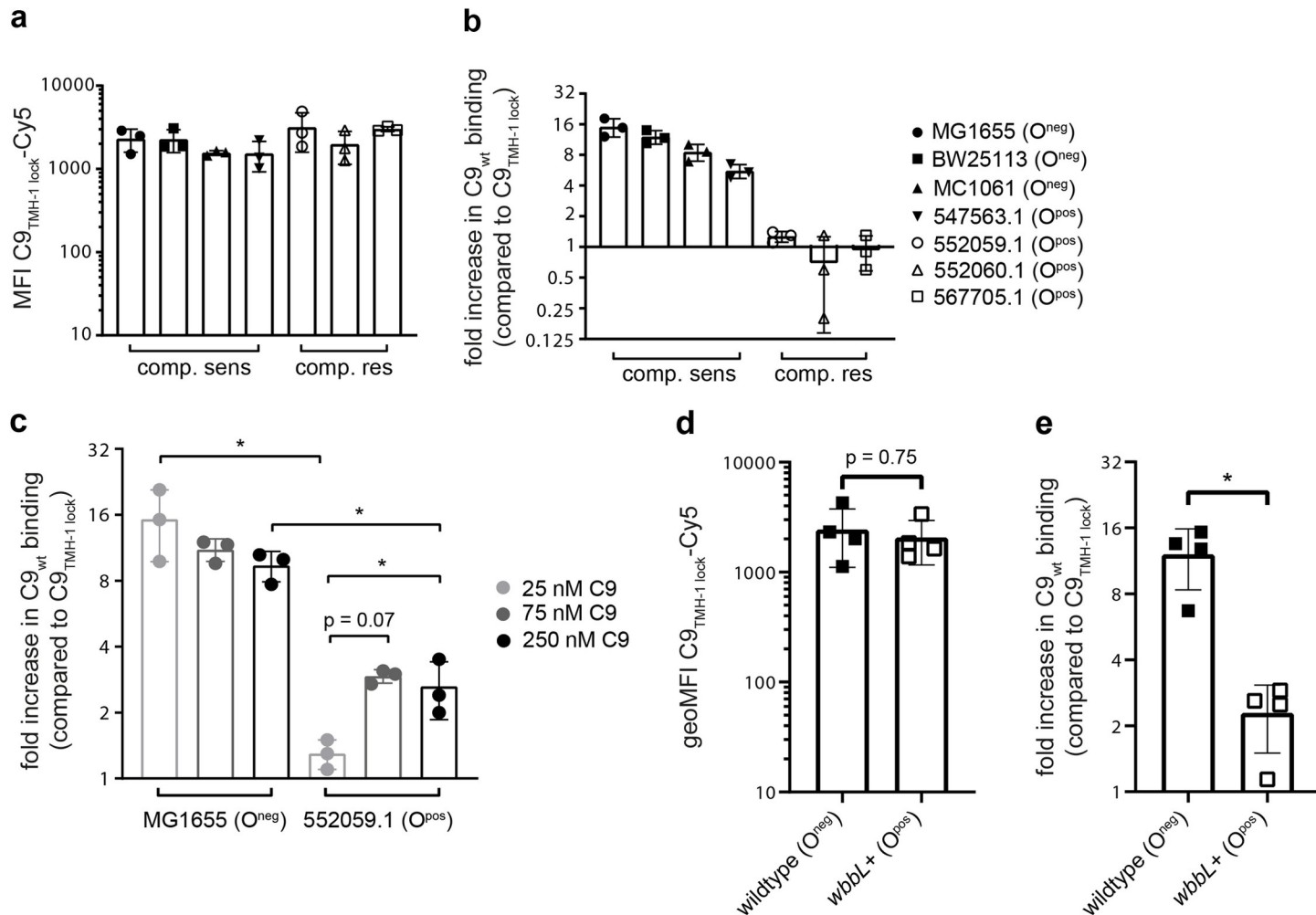

**Fig 6. Polymerization of C9 is impaired on complement-resistant *E. coli* that express LPS O-Ag.** a-c) Complement-resistant (552059.1, 552060.1 and 567705.1) and complement-sensitive (MG1655, BW25113, MC1061 and 547563.1) *E. coli* strains were incubated in 3% C9-depleted serum supplemented with a physiological concentration (= 25 nM) of Cy5-labelled $C9_{wt}$ or $C9_{TMH-1\ lock}$ for 30 minutes to measure C9 binding by flow cytometry. a) Binding of Cy5-labelled $C9_{TMH-1\ lock}$ to bacteria. b) The relative increase in $C9_{wt}$ binding compared to bacteria labelled with $C9_{TMH-1\ lock}$ was calculated as indication for C9 polymerization. c) The relative increase in $C9_{wt}$ binding compared to bacteria labelled with $C9_{TMH-1\ lock}$ was compared for complement-sensitive MG1655 and complement-resistant 552059.1 at different C9 concentrations. d-e) CGSC7740 wildtype and *wbbL*+ were incubated in 3% C9-depleted serum similarly as *E. coli* strains in a-c). d) Binding of Cy5-labelled $C9_{TMH-1\ lock}$ to bacteria. e) The relative increase in $C9_{wt}$ binding compared to bacteria labelled with $C9_{TMH-1\ lock}$ was calculated as indication for C9 polymerization. Flow cytometry data are represented by MFI values of the bacterial population. Data represent individual values with mean +/- SD of three (a, b and c) or four (d and e) independent experiments. Statistical analysis was done on log-transformed data ($^2$log for c and e, $^{10}$log for d) using a paired one-way ANOVA with Tukey's multiple comparisons' test (a-c) or a paired t-test (d and e). Significance was shown as $^*$ $p \leq 0.05$.

that it is important to efficiently damage the bacterial cell envelope and rapidly kill multiple Gram-negative bacterial strains and species.

Our findings suggest that bacteria are killed more rapidly when C9 polymerizes, both by MAC alone as well as in a serum environment. Previous reports have already suggested that the absence of C9 delays bacterial killing [37], and have correlated the presence of polymeric-C9 to bacterial killing [28,38]. Our study extends on these insights, since it provides direct evidence that polymerization of C9 enhances bacterial killing by using a system in which C9 can bind to C5b-8 without polymerizing. Although Spicer *et al.* had already demonstrated that locking the TMH-1 domain of C9 could prevent polymerization of C9 [30], we extend on

these insights by confirming that this does not affect binding of C9 to C5b-8 with direct binding experiments on *E. coli*. Apart from bacterial killing, our study also suggests that polymerization of C9 more efficiently damages both the OM and IM of the bacterial cell envelope. Polymerization of C9 strongly enhanced passage of small molecules through the OM and was required for passage of periplasmic proteins through the OM. Our findings therefore extend on previous reports that have suggested that polymeric-C9 is required for efficient cell envelope damage [27,28]. OM damage preceded IM damage, corresponding with our earlier findings [14]. Here, we add to these insights showing that OM damage precedes the loss of IM potential, which is a widely accepted characteristic of cell death.

Our data indicate that little polymeric-C9 already strongly enhanced bacterial killing and cell envelope damage. However, it remains unclear if a full polymeric-C9 ring of 18 C9 molecules is required. The aE11 antibody that is used for the detection of polymeric-C9 can also detect soluble MAC [32], which only contains up to 3 C9 molecules [39]. Although SDS-PAGE only revealed a single band for C9 polymers, it is still unknown if this band contains a mixture of C9 polymers of different sizes or if smaller C9 polymers are SDS-stable. Homologous pore-forming proteins of the MACPF/CDC family, such as suilysin and perforin, are known to form arc-shaped pores instead of ring-shaped pores [33,34]. It is therefore possible that arc-shaped MAC pores with only a few C9 molecules could be formed on bacteria and could enhance cell envelope damage and bacterial killing. Arc-shaped MAC pores have previously been observed on model-lipid membranes [17]. However, previous analyses on the bacterial OM with atomic force microscopy (AFM) primarily revealed the presence of ring-shaped MAC pores [14,15]. This suggests that even if arc-shaped MAC pores can be formed and have bactericidal capacity, it remains questionable if they are formed under physiological conditions.

Based on our results, we hypothesize that extensive OM damage by MAC pores is driving bacterial killing. Binding of C8 to C5b-7 allows passage of small molecules through the OM (**Fig 7-I**), but these lesions have minimal effect on the IM and bacterial viability. The presence of C9$_{\text{TMH-1 lock}}$ slightly increased damage to both the OM and IM. Based on the structure of monomeric-C9 [30], it is highly unlikely that the other TMH domain (TMH-2) of C9 can insert into the membrane when the TMH-1 domain is locked. However, we can never fully exclude that some C9$_{\text{TMH-1 lock}}$ still polymerizes. If there is some residual polymerization, it

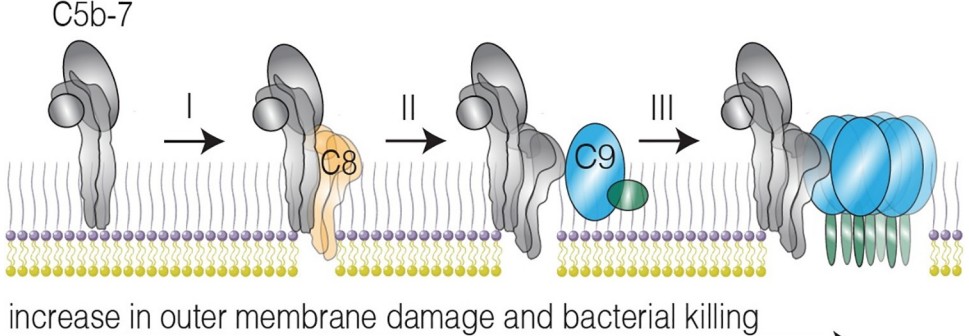

**Fig 7. Assembly of MAC pores enhances OM damage and bacterial killing.** Schematic overview how MAC pores assemble on *E. coli* and their effect on outer membrane damage and bacterial killing. I) C8 (orange) binds to membrane-anchored C5b-7 (grey) and subsequently inserts transmembrane β-hairpins into the bacterial outer membrane (OM), which causes small lesions in the OM. II) Binding of C9 (blue) to C5b-8 without polymerization of C9 slightly increases the OM damage and bacterial killing. III) C9 polymerizes and forms a transmembrane polymer, which drastically increases OM damage and rapidly kills bacteria.

seems likely that this would mainly result in arc-shaped and not ring-shaped MAC pores, which could explain why $C9_{TMH-1\ lock}$ primarily enhances passage of smaller molecules (such as DiOC2) through the OM compared to proteins such as $_{peri}$mCherry. However, our data do not show any direct evidence that $C9_{TMH-1\ lock}$ polymerized. We therefore hypothesize that binding of C9 to C5b-8 without polymerization of C9 could affect the stability by which C8 is inserted into the OM, which ultimately could affect cell envelope damage (**Fig 7-II**). Nonetheless, altogether our data highlight that polymerization of C9 drastically enhanced the damage of both the OM and IM damage and rapidly killed bacteria (**Fig 7-III**).

Our study primarily emphasizes that polymerization of C9 strongly enhances bacterial cell envelope damage and killing. How OM damage by MAC pores destabilizes the IM and kills bacteria remains unclear. The OM is an essential load-bearing membrane that confers stability to the bacterial cell [40]. The extent of OM damage by MAC pores could therefore determine whether a cell can cope with the increase in osmotic pressure. OM damage by MAC pores could even directly interfere with osmoregulation of the cell, as has been suggested for antimicrobial peptides that damage the OM of *E. coli* [41]. Moreover, since metabolism and growth-phase have been associated with sensitivity to killing by MAC pores [42,43], OM damage could also induce a stress response that affects the capacity of a cell to survive envelope stress. C9-depleted serum alone damaged the IM for one *E. coli* strain, even though the amount of C5b-8, indicated by the binding of $C9_{TMH-1\ lock}$, and OM damage was comparable to other *E. coli* strains. On the other hand, for *Klebsiella* strains IM damage appeared more dependent on polymerization of C9 than for *E. coli* strains. These differences between strains and species suggest that the capacity of a bacterium to cope with OM damage-related stress is a crucial determinant for bacterial killing by MAC pores. Further research looking directly at the cellular response of bacteria would be necessary to better understand how OM damage by MAC pores causes cell death.

Our data also suggest that an excess of C9 is required for efficient polymerization of C9. Joiner *et al.* already demonstrated that limiting the C9 concentration decreased the ratio of C9: C7 on the surface [38]. We further extend on these insights, because our data suggest that binding of C9 to C5b-8 is comparable or even favored over binding to C9 in the nascent polymeric-C9 ring ($C5b-9_x$). This suggests that the ratio between the generated C5b-8 and the amount of available C9 is important to ensure assembly of complete MAC pores. MacKay *et al.* have previously demonstrated that C9 binds comparably to C5b-8 or $C5b-9_1$ by adding C9 to *E. coli* labelled with C5b-8 at 0˚C [29]. By contrast, Parsons *et al.* recently suggested that binding and C9 to C5b-8 and subsequent insertion into the membrane is a kinetic bottleneck in the assembly of complete MAC pores [44]. However, Parsons *et al.* looked at MAC assembly in the presence of an excess of C9, which we here show to affect the assembly kinetics of MAC pores. Moreover, experiments in this study were done on model lipid-membranes that were not labelled with convertases, which also could have influenced the assembly of MAC pores [14]. Similar time-resolved AFM experiments on bacteria could provide more insight into the kinetics of the assembly of complete MAC pores on the bacterial surface.

Finally, our study highlights that polymerization of C9 is impaired on *E. coli* that resist MAC-dependent killing. Impaired polymerization of C9 correlated with the presence of LPS O-Ag in the tested *E. coli* strains. Moreover, restoring O-Ag expression in an O-Ag negative strain resulted in impaired polymerization of C9 and resistance to MAC-dependent killing. The presence and length of LPS O-Ag have previously been correlated with survival in serum [23]. Our data extend on these findings suggesting that the O-Ag results in complement-resistance by inhibiting polymerization of C9. Although LPS O-Ag modifications can also affect earlier steps in the complement cascade [9,10,45–47], this did not seem to play a major role as the amount of C5b-8 on the surface was comparable between complement-sensitive and

-resistant *E. coli*. How the O-Ag affects polymerization of C9 remains unclear. The O-Ag could affect the distance of the nascent MAC to the OM and thereby the accessibility of hydrophobic patches in the OM [8]. Our previous study demonstrated that C9 is not efficiently inserted into the OM of the same complement-resistant *E. coli* used in this study [15], because C9 was more sensitive to trypsin digestion compared to complement-sensitive *E. coli*. We therefore hypothesize that the presence of O-Ag results in MAC assembly further away from the OM, which prevents C9 from inserting into the OM and forming a polymeric-C9 ring. Future experiments looking directly at the distance of MAC components to the OM could help further understand how the O-Ag affects MAC assembly on complement-resistant *E. coli*.

In conclusion, our study provides insight into how MAC pores damage the bacterial cell envelope and kill Gram-negative bacteria. Moreover, our study highlights how bacteria resist killing by MAC pores. These fundamental insights are important to understand how the complement system prevents infections and how bacteria escape killing by the immune system. Ultimately, these insights may guide the development of future immune therapy against bacteria.

## Materials and methods

### Serum and complement proteins

Serum depleted of complement components C5, C8 or C9 was obtained from Complement Technology. Serum was thawed and aliquoted, but not subjected to any further freeze-thaw cycles. Preassembled C5b6 (pC5b6) and C8 were also obtained from Complement Technology. His-tagged complement components C5, C6 and C7 were expressed in HEK293E cells at U-Protein Express as described previously [15]. OmCI was produced in HEK293E cells at U-Protein Express as well and purified as described before [48]. Monoclonal mouse-anti polymeric-C9 (aE11, kindly provided by T. Mollness and P. Garred) was randomly labelled with NHS-Alexa Fluor AF488 (ThermoFisher) according to manufacturer's protocol. Lysozyme was obtained from Raybio and recombinant type IIa secreted phospholipase 2A (PLA) was kindly provided by Gérard Lambeau [49].

### Expression and purification of C9

C9$_{wt}$ and C9$_{TMH-1\ lock}$ (F262C V405C) were cloned into the vector pcDNA34 (Thermo Fisher Scientific) that was modified with a NheI/NotI multiple cloning site. This vector contains a cystatin-S signal peptide and was further modified to encode for the expression of a C-terminal AAA-3x(GGGGS)-LPETGG-HHHHHH tag. gBlocks (Integrated DNA Technologies) containing codon optimized C9 sequences were cloned via Gibson assembly into the NheI/NotI digested pcDNA34 and transformed into Top10F' *E. coli*. After verification of the correct sequence, the plasmids were used to transfect EXPI293F cells. EXPI293F cells were grown in EXPI293 medium (Life Technologies) in culture filter cap Erlenmeyer bottles (Corning) on a rotation platform (125 rotations/min) at 37°C, 8% CO$_2$. One day before transfection, cells were diluted to 2x10$^6$ cells/ml. The next day, cells were diluted to 2x10$^6$ cells/ml using SFM4Transfx-293 medium, containing UltraGlutamine I (VWR International) prior to transfection using PEI (Polyethylenimine HCl MAX; Polysciences). 0.5 μg DNA/ml cells, containing 50% empty (dummy) vector, was added to Opti-MEM (1:10 of total volume; Gibco) and gently mixed. After adding 1 μg/ml PEI in a PEI/DNA (w/w) ratio of 5:1, the mixture was incubated at room temperature for 20 min and added dropwise to cells while manually rotating the culture flask. After 3.5 hours, 1 mM valproic acid (Sigma) was added. After 5 days of expression, the cell supernatant was collected by centrifugation and filtration (0.45 μm). Cell supernatant was diafiltrated over a 30 kDa membrane on a Quixstand (GE healthcare) to Tris/NaCl

buffer (50 mM Tris/500 mM NaCl at pH 8.0). Proteins were finally loaded on a $Ni^{2+}$-loaded HiTrap HP Chelating column (GE healthcare) in Tris/NaCl buffer supplemented with 40 mM imidazole and eluted with 150 mM imidazole. Final purification was done by size-exclusion chromatography (SEC) on a Superdex 200 Increase column (GE Healthcare) on an Akta Explorer (GE Healthcare) with PBS. The concentration of proteins was determined by measuring absorbance at 280 nm and verified by SDS-PAGE.

## Site-specific fluorescent labelling of MAC components

C6 and C9 were labelled with fluorescent probes as described previously [14,15]. 50 μM of protein with C-terminal LPETGG-His tag was incubated with 25 μM His-tagged sortase-A7+ [50] and 1 mM GGG-substrate in Tris/NaCl buffer (50 mM Tris/300 mM NaCl at pH 7.8) for two hours at 4˚C. GGGK-FITC (Isogen Life Science) was used for C6-LPETG-G-His and GGGK-azide (Genscript) for C9-LPETGG-His. Sortagged proteins were purified on a HisTrap FF column (GE Healthcare), which captures protein that was not sortagged and still contains a His-tag. FITC-labelled C6 was directly purified by SEC on a Superdex 200 Increase column on the Akta Explorer with PBS. GGG-azide labelled proteins were concentrated to 25 μM on a 30 kDa Amicon Tube (Merck Millipore) in Tris/NaCl buffer and next labelled with 100 μM DBCO-Cy5 (Sigma Aldrich) via copper-free click chemistry for 3h at 4˚C. Finally, Cy5-labelled proteins were also purified by SEC on a Superdex 200 Increase column with PBS. Labelling of the proteins was monitored during SEC by measuring absorbance at 280 nm (protein), 488 nm (FITC) and 633 nm (Cy5) nm and finally verified by SDS-PAGE by measuring in-gel fluorescence with LAS4000 Imagequant (GE Healthcare).

## Bacterial strains

Unless otherwise specified, the common laboratory *E. coli* strain MG1655 was used in our experiments. For experiments where leakage of periplasmic mCherry (perimCherry) was measured, MG1655 was used transformed with pPerimCh containing a constitutively expressed perimCherry previously used in [14]. Other laboratory *E. coli* strains that were used in this study included BW25113 and MC1061. Clinical isolates, namely *E. coli* 547563.1, 552059.1, 552060.1, 567705.1, *Klebsiella variicola* 402 and *Klebsiella pneumoniae* 567880.1, were obtained from the clinical Medical Microbiology department at the University Medical Center Utrecht. CGSC7740 wildtype and *wbbL+* were kindly provided by Benjamin Sellner (Biozentrum, University of Basel). CGSC7740 *wbbL+* was constructed by replacing the IS5-element present in the *wbbL+* gene with a *sacB-kan* cassette to select for kanamycin-resistance. The *sacB-kan* cassette was then replaced with wildtype *wbbL* without the IS5-element and selected by counter selection on sucrose.

## Bacterial growth

For all experiments, bacteria were plated on Lysogeny Broth (LB) agar plates. Single colonies were picked and grown overnight at 37˚C in LB medium. For MG1655 transformed with pPerimCh, LB was supplemented with 100 μg/ml ampicillin. The next day, subcultures were grown by diluting at least 1/30 and these were grown to mid-log phase (OD600 between 0.4–0.6). Once grown to mid-log phase, bacteria were washed by centrifugation three times (11000 rcf for 2 minutes) and resuspended to OD 1.0 (~1 x $10^9$ bacteria/ml) in RPMI (Gibco) + 0.05% human serum albumin (HSA, Sanquin).

## Complement labelling and serum bactericidal assays

For MAC-specific bactericidal assays, bacteria were labelled with C5b-7 as described previously [15]. In short, bacteria (~1 x $10^8$ bacteria/ml) were incubated with 10% C8-depleted serum (v/v) for 30 minutes at 37°C, washed three times and resuspended in RPMI-HSA. C5b-7 labelled bacteria (~5 x $10^7$ bacteria/ml) were incubated for 30 minutes at 37°C with 10 nM C8 and 20 nM C9, unless stated differently. When C9 was reduced with 10 mM dithiothreitol (DTT), 20 nM C8 was added to bacteria (~5 x $10^7$ bacteria/ml) 15 minutes before C9 was added to allow binding of C8 to C5b-7 at RT. Washing steps were done by pelleting bacteria at 11,000 rcf for 2 minutes and washing with RPMI-HSA. For serum bactericidal assays, bacteria (~5 x $10^7$ bacteria/ml) were incubated with 3% C9-depleted serum (v/v) supplemented with physiological concentrations of C9 (100% serum ± 1 μM) for 30 minutes at 37°C, unless stated differently. KP880.1 was incubated in 10% C9-depleted serum (v/v) with the corresponding physiological concentration of C9. Blocking of C5 conversion in serum was done with 25 μg/ml OmCI as final concentration. For assays where nisin (Handary, SA, Brussels) was added, 3 μg/ml was used as final concentration.

## Flow cytometry

Complement-labelled bacteria (~5 x $10^7$ bacteria/ml) were incubated with 2.5 μM of Sytox Blue Dead Cell stain (Thermofisher). Samples were diluted to ~2.5 x $10^6$ bacteria/ml in RPMI-HSA and subsequently analyzed on a MACSquant VYB (Miltenyi Biotech) for Sytox and $_{peri}$mCherry fluorescence. Polymeric-C9 deposition was measured by incubating bacteria (2.5 x $10^7$ bacteria/ml) with 6 μg/ml monoclonal AF488 labelled mouse-anti polymeric-C9 (aE11) for 30 minutes at 4°C. For C9 binding in serum, bacteria were stained with 1 μM Syto9 (Thermofisher) to exclude serum noise events. Samples were next diluted to ~2.5 x $10^6$ bacteria/ml in 1.1% paraformaldehyde (v/v) and subsequently analyzed on the BD FACSVerse flow cytometer for Cy5 and aE11-AF488 or Syto9 fluorescence. Flow cytometry data was analysed in FlowJo V.10. Bacteria were gated on forward scatter and side scatter. In serum, an additional trigger was placed on Syto9 fluorescence. Sytox positive cells were gated such that the buffer only control had <1% positive cells.

## Polymeric-C9 detection by SDS-PAGE

Bacterial labelled with complement components were collected by spinning bacteria down 11,000 rcf for 2 minutes and subsequently washing cell pellets twice in RPMI-HSA. Cell pellets were resuspended and diluted 1:1 in SDS sample buffer (0.1M Tris (pH 6.8), 39% glycerol (v/v), 0.6% SDS (m/v) and bromophenol blue) supplemented with 50 mg/ml DTT and placed at 95°C for 5 minutes. Samples were run on a 4–12% Bis-Tris gradient gel (Invitrogen) for 75 minutes at 200V. Gels were imaged for 10 minutes with increments of 30 seconds on the LAS4000 Imagequant (GE Healthcare) for in-gel Cy5 fluorescence. Monomeric-C9 and polymeric-C9 were distinguished by size, since monomeric-C9 runs at 63 kDa and polymeric-C9 is retained in the comb of the gel.

## Bacterial viability assay

Bacteria were treated with MAC components or serum as described above. Next, colony forming units (CFU) were determined by making serial dilutions in PBS (100, 1.000, 10.000 and 100.000-fold). Serial dilutions were plated in duplicate on LB agar plates and incubated overnight at 37°C. The next day, colonies were counted and the corresponding concentration of CFU/ml was calculated.

## Multi-well fluorescence assays

Bacteria (~5 x $10^7$ bacteria/ml) added to RPMI-HSA supplemented with 1 μM of Sytox Green Dead Cell stain (Thermofisher) or 30 μM $DiOC_2$ (PromoCell). Bacteria were next incubated with MAC components, serum and/or nisin as described above. Fluorescence was measured every 60 seconds on a Clariostar platereader (BMG labtech). Sytox green fluorescence was measured using an excitation wavelength of 484–15 nm and emission wavelength of 527–20. $DiOC_2$ fluorescence was measured using an excitation wavelength 484–15 nm and emission wavelength of 527–20 (green) and 650–24 (red). For $DiOC_2$, the red fluorescence was divided by the green fluorescence to determine the red:green fluorescence ratio. $t_{1/2peak}$ for $DiOC_2$ was interpolated at half the value of the peak, which was calculated by subtracting the background ratio value at t = 0 from the peak ratio value and dividing by two. $t_{1/2maximum}$ for Sytox was interpolated at half the value of the maximum Sytox value, which was calculated by subtracting the fluorescence value at t = 0 from the maximum fluorescence value and dividing by two.

## LPS O-antigen Silver staining

*E. coli* strains were typed for LPS O-Ag by Silver staining after SDS-PAGE based on [51,52]. In short, bacteria were scraped from blood agars plates, resuspended in PBS and incubated at 56˚C for 60 minutes. Cell pellets were next deproteinated with 400 μg/ml proteinase K for 90 minutes and diluted in 2x Laemli buffer with 0.7 M beta-mercaptoethanol. Cell pellets were run on a 4–12% BisTris gel as described above and fixed overnight in fixing buffer (40% ethanol (v/v) + 4% (v/v) glacial acetic acid). The gel was oxidized for 5 minutes in fixing buffer supplemented with 0.6% (m/v) periodic acid. The gel was then stained for 15 minutes with freshly prepared 0.3% silver nitrate in 0.125 M sodium hydroxide and 0.3% ammonium hydroxide (v/v). Finally, the gel was developed for 7 minutes in developer solution (0.25% citric acid (m/v) + and 0.2% formaldehyde (v/v)). In between steps, the gel was washed three times with MilliQ. Lipid A and LPS core were distinguished from LPS O-Ag based on size.

## Data analysis and statistical testing

Unless stated otherwise, graphs are comprised of at least three biological replicates. Statistical analyses were performed in GraphPad Prism 8 and are further specified in the figure legends.

## Supporting information

**S1 Fig. Labelling C9 with Cy5 and validating that locking the TMH-1 domain impairs polymerizing capacity.** a) SDS-PAGE of a concentration range of Cy5-labelled $C9_{wt}$ and $C9_{TMH-1\ lock}$ (in μg/ml). Both InstantBlue staining (top) and in-gel Cy5 fluorescence (bottom) were shown. b) Sheep erythrocytes (4%) labelled with rabbit anti-sheep IgM were incubated in 2% C9-depleted serum for 30 minutes. Next, erythrocytes were washed and incubated with a concentration range of $C9_{wt}$ or $C9_{TMH-1\ lock}$ in the presence or absence of 10 mM DTT. The percentage of lysed erythrocytes was calculated by adding MilliQ as 100% lysis or buffer as 0% lysis control after incubation with C9-depleted serum. c) 20 nM pC5b6, 20 nM C7, 20 nM C8 was incubated with 100 nM Cy5-labelled $C9_{wt}$ or $C9_{TMH-1\ lock}$. SDS-PAGE was done to distinguish monomeric-C9 from polymeric-C9 by in-gel Cy5 fluorescence. (d) C6-FITC binding to *E. coli* MG1655 measured by flow cytometry. Bacteria were labelled with C5b-7 by incubating them in 10% C8-depleted serum supplemented 12 μg/ml C6-FITC with for 30 minutes. Bacteria were washed and next incubated with 10 nM C8 for 15 minutes. Finally, 20 nM of Cy5-labelled $C9_{wt}$ or $C9_{TMH-1\ lock}$ was added in the presence or absence of 10 mM DTT for 30 minutes. Flow cytometry data are represented by geoMFI values of the bacterial population.

Data represent mean +/- SD (b) or individual values with mean +/- SD (d) of three independent experiments. Statistical analysis was performed using a paired one-way ANOVA with Tukey's multiple comparisons' test. For d, data were [10]log-transformed. Significance was shown as $*$ p ≤ 0.05, $**$ p ≤ 0.005, $****$ $**$ p ≤ 0.0001.
(TIF)

**S2 Fig. Unlocking TMH-1 domain after C9 has bound C5b-8 does not affect cell envelope damage.** *E. coli* MG1655 was labelled with C5b-7 by incubating in 10% C8-depleted serum for 30 minutes. Bacteria were washed and next incubated with 10 nM C8 and 50 nM C9$_{wt}$ or C9$_{TMH-1 lock}$ for 15 minutes. Next, bacteria were washed (@) and incubated with buffer or 5 mM DTT for 30 minutes. Bacteria were analyzed by flow cytometry for Sytox influx to determine inner membrane damage (a) and leakage of periplasmic mCherry ($_{peri}$mCherry) for outer membrane damage (b). Flow cytometry data are represented by relative geoMFI values of the bacterial population compared to bacteria in buffer. Data represent individual values with mean +/- SD of three independent experiments.
(TIF)

**S3 Fig. Polymerization of C9 enhances nisin influx.** *E. coli* MG1655 was labelled with C5b-7 by incubating in 10% C8-depleted serum for 30 minutes. Bacteria were washed and next incubated with 10 nM C8 and 20 nM of C9$_{wt}$ or C9$_{TMH-1 lock}$ supplemented with or without 3 µg/ml nisin to measure passage of nisin through the OM. Nisin influx was determined by measuring Sytox influx over time in a multi-well plate-reader assay. The graph shows one representative experiment that has been repeated at least three times.
(TIF)

**S4 Fig. Increasing the amount of surface-bound C5b-8 impairs polymerization of C9.** *E. coli* MG1655 was labelled with C5b-7 by incubating in 10% C8-depleted serum for 30 minutes. Bacteria were washed and next incubated with a concentration range of C8 and 3 nM (a,b,d) or 1 nM (c) Cy5-labelled C9$_{wt}$ or C9$_{TMH-1 lock}$ for 30 minutes. a) Binding of Cy5-labelled C9$_{wt}$ or C9$_{TMH-1 lock}$ to bacteria measured by flow cytometry. b) The relative increase in C9$_{wt}$ binding compared to bacteria labelled with C9$_{TMH-1 lock}$ was calculated as indication for C9 polymerization. c) periplasmic mCherry ($_{peri}$mCherry) leakage was measured after 30 minutes by flow cytometry and represented as relative $_{peri}$mCherry fluorescence compared to t = 0. d) Bacterial cell pellets were analyzed by SDS-PAGE for in-gel fluorescence of Cy5-labelled C9$_{wt}$ to distinguish monomeric-C9 from polymeric-C9. Flow cytometry data are represented by geoMFI values of the bacterial population. Data represent mean +/- SD of three independent experiments. The SDS-PAGE image is representative for at least three independent experiments.
(TIF)

**S5 Fig. C9$_{TMH-1 lock}$ increases passage of lysozyme and phospholipase IIa-2A through the outer membrane compared to C5b-8 alone.** *E. coli* MG1655 was labelled with C5b-7 by incubating in 10% C8-depleted serum for 30 minutes. Bacteria were washed and next incubated with 10 nM C8 and 20 nM of C9$_{wt}$ or C9$_{TMH-1 lock}$. a) Bacterial viability was determined at different time points by counting colony forming units (CFU's) and calculating the proportion of viable cells compared to t = 0. b) *E. coli* MG1655 bacteria were incubated with 10% C8 depleted serum for 30 minutes. After washing, bacteria were incubated with buffer, 10 nM C8 or 10 nM C8 + 20 nM C9 in the presence of 5 µg/ml lysozyme (lys) or 0.3 µg/ml recombinant type IIa secreted phospholipase 2A (PLA). Bacterial viability was determined after 90 minutes of incubation by counting CFU's and calculating the proportion of viable cells compared to t = 0. The horizontal dotted line represents the detection limit of the assay. Data represent mean +/- SD

(b) or individual values with mean +/- SD (a) of three independent experiments. Statistical analysis was done on $^{10}$log-transformed data (b) using a paired one-way ANOVA with Tukey's multiple comparisons' test. Significance was shown as $^{**}$ p $\leq$ 0.005.
(TIF)

**S6 Fig. Polymerization enhances cell envelope damage on multiple Gram-negative strains.** *E. coli* strains (BW25113, MC1061, 547563.1) and *K. variicola* 402 were incubated in 3% C9-depleted serum supplemented with a physiological concentration (= 25 nM) of C9$_{wt}$ or C9$_{TMH-1 lock}$ for 90 minutes. *K. pneumoniae* 567880.1 was incubated in 10% C9-depleted serum supplemented with 80 nM C9$_{wt}$ or C9$_{TMH-1 lock}$. a) OM damage was measured by DiOC$_2$ influx, which was determined by the shift in red:green fluorescence ratio over time in a multiwell plate-reader assay. b) IM damage was measured by Sytox influx over time in a multi-well plate-reader assay. Graphs represent one representative experiment that has been repeated at least three times.
(TIF)

**S7 Fig. LPS O-Antigen expression confers complement-resistance.** a) Complement-resistant *E. coli* 552059.1 was incubated in 3% C9-depleted serum supplemented with 25, 75 or 250 nM C9$_{wt}$, C9$_{TMH-1 lock}$ or 25 μg/ml OmCI. Bacterial viability was determined by counting colony forming units (CFU's) per ml. The horizontal dotted line represents the detection limit of the assay. b) Complement-sensitive (MG1655, BW25113, MC1061 and 547563.1) and complement-resistant (552059.1, 552060.1, 567705.1) *E. coli* strains were typed for the presence of LPS O-Antigen (O-Ag) via Silver staining. LPS-core was distinguished from LPS O-Ag based on size. c) CGSC7740 wildtype (wt) and *wbbL*+ were typed for the presence of LPS O-Ag via Silver staining as done for b. d) CGSC7740 wt and *wbbL*+ were incubated in 5% normal human serum (NHS) supplemented with and without 25 μg/ml OmCI. Bacterial viability was determined by counting CFU's and calculating the proportion of viable cells compared to t = 0. The horizontal dotted line represents the detection limit of the assay. SDS-PAGE images are representative for at least two independent experiments. Data represent individual values with mean +/- SD of three independent experiments (b and d).
(TIF)

# Acknowledgments

The authors would like to acknowledge Benjamin Sellner and Urs Jenal for providing the CGSC7740 wildtype and *wbbL*+ strain. The authors would like to acknowledge Lisanne de Vor and Myrthe Reiche for critically reading the manuscript.

# Author Contributions

**Conceptualization:** Dennis J. Doorduijn, Suzan H. M. Rooijakkers, Bart W. Bardoel.

**Data curation:** Dennis J. Doorduijn.

**Formal analysis:** Dennis J. Doorduijn, Maartje Ruyken, Piet C. Aerts.

**Funding acquisition:** Suzan H. M. Rooijakkers.

**Methodology:** Dennis J. Doorduijn, Dani A. C. Heesterbeek, Carla J. C. de Haas, Daphne A. C. Stapels, Bart W. Bardoel.

**Project administration:** Suzan H. M. Rooijakkers, Bart W. Bardoel.

**Resources:** Carla J. C. de Haas, Piet C. Aerts.

**Supervision:** Suzan H. M. Rooijakkers, Bart W. Bardoel.

**Validation:** Maartje Ruyken.

**Writing – original draft:** Dennis J. Doorduijn, Bart W. Bardoel.

**Writing – review & editing:** Dennis J. Doorduijn, Dani A. C. Heesterbeek, Daphne A. C. Stapels, Suzan H. M. Rooijakkers, Bart W. Bardoel.

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
