## [Decision Letter · Decision Letter 0]

13 Jul 2021

Dear Dr Bardoel,

Thank you very much for submitting your manuscript "Polymerization of C9 enhances bacterial cell envelope damage and killing by membrane attack complex pores" for consideration at PLOS Pathogens. As with all papers reviewed by the journal, your manuscript was reviewed by members of the editorial board and by several independent reviewers. In light of the reviews (below this email), we would like to invite the resubmission of a significantly-revised version that takes into account the reviewers' comments.

Thank you for submitting your manuscript to PLOS Pathogens. We have now finally received the reviews, which unfortunately were a bit late. As you will see the comments were many and ranged broadly.

As the reviewers found also merit in your work and manuscript, we would like to invite a thoroughly revised manuscript for a second review. However, we can not guarantee publication, unless the requirement by the reviewers are met appropriately. I sincerely hope that you will find the expert comments helpful.

We cannot make any decision about publication until we have seen the revised manuscript and your response to the reviewers' comments. Your revised manuscript is also likely to be sent to reviewers for further evaluation.

Sincerely,

Seppo Meri

Guest Editor

PLOS Pathogens

Christoph Tang

Section Editor

PLOS Pathogens

Kasturi Haldar

Editor-in-Chief

PLOS Pathogens

orcid.org/0000-0001-5065-158X

Michael Malim

Editor-in-Chief

PLOS Pathogens

orcid.org/0000-0002-7699-2064

Dear Authors

Thank you for submitting your manuscript to PLOS Pathogens. We have now finally received the reviews, which unfortunately were a bit late. As you will see the comments were many and ranged broadly.

As the reviewers found also merit in your work and manuscript, we would like to invite a thoroughly revised manuscript for a second review. However, we can not guarantee publication, unless the requirement by the reviewers are met appropriately. I sincerely hope that you will find the expert comments helpful.

With best regards

Seppo Meri

Guest Editor

PLOS Pathogens

Reviewer's Responses to Questions

**Part I - Summary**

Reviewer #1: In this manuscript the authors make use of the ingenious “locked C9” reported by the Dunstone group in 2018 to explore whether MAC-mediated bacterial killing requires C9 polymerisation. The manuscript is very well written and easy to read – for which this reviewer is grateful. I do have some concerns about the presentation and interpretation of the interesting data; these are listed below.

Reviewer #2: This is a very well written paper that uses innovative tools to evaluate the role of C9 in killing of gram-negatives. This paper unifies several older observations and also lends new insights into the role of C9 in killing gram negatives. Differences across strains and species with respect to how MAC is assembled and inserted have been elucidated. The experiments all have stringent controls. The data have been presented very clearly and the discussion is excellent.

Reviewer #3: This work closely follows on from a previous Plos pathogens paper that describes how the MAC facilitates pore formation of the outer membrane of E. coli and delivery of other factors that damage cell wall/membrane integrity. This particular manuscript explores whether the MAC needs to form a full pore or not and how incomplete pore formation affects function although some of the research described as kinetics contradicts published SM, time-resolved work. There is also exploration of how the MAC might be inhibited in complement resistant E. coli. Overall there is an interesting set of experiments but it is difficult to explain as the regain of function of the lock mutant in reducing conditions is not perfect and the DTT does affect the wtC9 function probably due to the partial loss of the normal disulphide bonds required for folding and stability of C9. This data needs to be presented and interpreted in a more upfront manner with the interpretation that the mutant is not completely inactive.

**Part II – Major Issues: Key Experiments Required for Acceptance**

Reviewer #1: In the abstract and elsewhere the authors state that there has been, prior to the availability of the “locked C9” reagent, “no robust way to specifically prevent polymerization of C9”. I refer the authors to the excellent 1994 paper from McKay and Dankert that uses temperature shifts to do exactly this in the context of bacterial killing. No doubt, the current work builds on this observation but they should fully acknowledge it.

At the risk of being pedantic, the description of the MAC in lines 57/58 does not match the description in the references used; in these Bubeck’s group show clearly that the pore is composed of 22 staves, 18 from C9 and one each from C6, C7, C8a and C8b. Similarly pedantic – but important I think; Spicer demonstrated rather than suggested the lock C9 (line 71).

The first dataset explores the impact of C9 polymerisation. Although I usually hate cartoons in figures, 1a and 1b are nice and likely to be of help to non-experts. Supp1b shows minimal (but not zero) ShE lysis using the locked C9; the Spicer paper found no haemolytic activity in the locked protein and the model proposed would suggest no effect – do the authors think that this is real? Figures c/d show binding of C9wt and C9lock to C5b-8, replicating what was shown in Spicer et al, albeit in the context of E coli, clarify that this is confirmatory. Figures 1e and S1c show C9 monomer/polymer in slot blots with no markers or other orientations. Here and elsewhere I strongly urge that full gels/blots with clear markers are used. Next, the authors test bacterial lysis using a neat method they have established; Fig 2a shows weak but significant activity in the C9lock – again, given the model proposed, any activity is surprising. Do the authors have an explanation of this? The impact of releasing the lock on lysis and viability is clear in 2b/c. The dose-dependent inhibition of poly-C9 formation shown in fig 2d is intriguing. Spicer showed very similar data in a different assay system and reached a similar conclusion – that the C9lock could bind and stall polymerisation at different stages; the current data though do beg the question of how many unfolded C9 molecules are needed for the ae11 epitope – given that the mAb detects TCC (2 or 3 C9s) likely not many! Do lower Mr ae11-stained bands appear? Are these sub-pore complexes SDS-labile? The bacterial lysis kinetics in figure 2e look very different – more of a threshold effect and kicking in at equimolar and above – state that this shows “that only very few C9 polymers are required to damage the IM”; however, it could equally show that even complexes with only a few C9s are lytic, the converse of the hypothesis.

Here and elsewhere I wonder why the authors have not washed after binding C9lock then unlocking in order to properly explore the impact of insertion of that first C9 – seems an obvious addition to many of the experiments that would distinguish effects of C5b-8, C5b-8-C9lock and C5b-8-C9unlock.....

OM lysis is next explored; 3a/b both show that C5b-8/C9wt causes OM damage while C5b-8/C9lock does not (as expected); a surprising omission is testing the impact of lock release – binding C9lock, washing and unlocking to test impact of C5b-8/C9 (single C9). Fig 3c shows a surprising residual activity in C9lock – state that this “suggests that binding of C9 to C5b-8 in the absence of polymerization slightly increases OM damage”. Again, this doesn’t fit the model – what might the mechanism be?

Next the authors explore binding and polymerisation of C9wt and C9lock to C5b-8-coated bugs. By titrating the C9s they show an initial similar increase in binding with dose up to an inflexion point where the C9lock is saturated but C9wt continues to bind. Only above this inflexion point is poly-C9 seen in bacterial pellets – although monomeric C9 is also only seen above this point and no data are shown for C9lock, presumably because neither mono- or poly-C9 can be detected; the data in 4d do not support their statement that SDS-PAGE shows only monomeric C9 at lower concentrations. The authors suggest this indicates that “C9 starts polymerizing when all C5b-8 complexes have bound one C9 molecule”; they note that this interpretation directly contradicts the Parsons study which shows, albeit in an artificial membrane system, that capture of the first C9 is rate-limiting and thereafter further C9 recruitment is rapid. This is an important clarification but I’m not sure that the data fully support it. It seems equally logical to suggest that at low doses all C9 offered is bound, either directly to C5b-8 (only that for C9lock) or both to C5b-8 and C5b-9 (for C9wt); when the C5b-8 sites are saturated the curves diverge, as is seen. Neither poly- nor mono-C9 binding is detected at the low C9 inputs as shown in fig 4d (note comment above regarding slotblots). The strong statement that “these data suggest that polymerization of C9 is rate-limiting in the assembly of complete MAC pores on bacteria” is not well supported by the data – kinetic analyses of the sort described in the Parsons study are likely needed.

Figure 5 shows killing of bacteria in C9-depleted serum supplemented with C9wt of C9lock to explore the requirement for C9 polymerisation for MAC effects on bacteria. These straightforward experiments confirm that polymerisation of C9 in the MAC is essential for efficient bacterial lysis/killing. The findings are complicated by the observation that killing in the presence of C9lock is delayed rather than prevented; they suggest that this is due to continued MAC deposition (supported by OmCI block) and/or presence of other bactericidal serum components. It would be interesting to confirm that there is no “lock-release” in these prolonged serum incubations that might enable polymerisation. The final dataset extends the key observation of requirement for polymerisation for efficient killing to other bacteria and also makes the interesting observation that C9 polymerisation (not C5b-8 formation or first C9 binding) is limited on resistant strains.

Reviewer #2: none

Reviewer #3: In this sense there is a lot of conceptual overlap with the previous Plos pathogens paper (ref 31 in this manuscript) which shows delivery of other factors. An explanation in the introduction from the authors about then potential overlap and how this is original research would be appreciated. I think that it is really important to put this upfront in the paper as there is a sense that there is redundancy in this manuscript with ref 31.

Figure 1: I find the diagram in figure 1A is confusing as it depicts C9 as binding to C5b-8 and then moving away from C5b-8 as a monomer. The monomer changes shape and then recruits a second monomer of C9. This is not actually how the Membrane attack complex forms and would confuse all readers regardless of whether they are experts or not. This needs to be fixed. The term polyC9 should only be used for the unusual characteristic of human C9 whereby it can form a homomeric ring that looks essentially like a pore but is really a non-physiological phenomenon. This figure needs major work.

Lines 103-5: The author's only use bar graphs to demonstrate the loss of activity for the disulfide trap mutant compared to wtC9. Whilst it might appear to be a clean cut binary style observation that there is either no activity in non-reducing conditions and activity in reducing conditions, when the original paper (Spicer et al.) is looked at, there is a lower LD50 for the reduced disulphide C9 compared to wtC9. Under no circumstances is it acceptable to have only bar graphs of activity for cytolytic protein activity. The original titration based activity profiles (done as replicates) should be presented for this type of data. This will provide a more qualitative view of the activity profile of the proteins being used and is a more honest way of showing the regain of function in the given reducing conditions. Curves should go to a relevantly high concentration of C9 as it is not uncommon for there to be some residual activity. The authors show a loss of wtC9 activity in the presence of DTT but this is not observed in the original Spicer paper so, again, the raw titration based curves would be important here.

Lines 113-123 and Figure 1c, e: I find that the use of the log scale on the Y-axis of Fig 1c dilutes the visual perception of the differences in binding of the trap mutant with and without DTT. In effect the binding of the lock mutant in the DTT is half of wt C9 in DTT. I would say from this data that the C9 lock mutant is half as good at binding/polymerising to the C5b-8 complex and maybe this should be stated as such. This interpretation is also supported by Fig 1e with a much smaller “polyC9” band which was correctly interpreted by the authors. What is missing is also the binding of aE11 to the reduced C9 lock mutant in Figure 1f to quantitate the formation of polymerised C9 and a more honest, quantitated statement of the amount of C9 polymerisation in the reduced form of the C9 lock mutant. I.e. it is not a perfect regain of wt-like activity and should be interpreted as such.

Figure 2a,b: It would be easier for the reader to interpret the experiments correctly if you combined Figures 2 a and b.

Lines 142-144, figure 2d,e: I find this to be a really important observation that could be expanded. Is it possible to interpret that arcs are able to allow the damage of the inner membrane at this stage of the manuscript? And refer to the background of data of arcs which is quite extensive for the MACPF/CDC family.

Lines 149 - 55, Figure 3a: the y-axis is not a linear-scale not a log scale. Not consistent with the other experiments. Is this experiment compared to E. coli not expressing peri-mCherry for background scattering as the baseline? Overall this is a weak observation but that is fine. The weak change may suggest that even with wtC9 there are only a few full pores that are capable of passaging mCherry as opposed to the passage of nisin and this should be noted in the results interpretation. Statistics need to be assigned to state it this is a significant observation or not as it looks like it is not. Maybe present the nisin work which is more significant before the mCherry work and state clearly the potential for arc formation as a possible explanation.

Lines 168-72: Given that there is potential background pore formation for the C9 lock mutant which should be explained in a more upfront way (i.e. Figure 1c), then a better/alternative explanation for Fig 3c is that there is still some background MAC formation that allows the passage of DiOC2. Again I think this is perfectly logical. There is also evidence of leaking membranes with the C5b-8 complex which may explain the black triangle curve.

Discussion: There definitely needs to be discussion about the role of arcs or incomplete MAC pores that can allow some passage of some molecules (DiOC2 and nisin) but not mCherry. And this should be reflected in the model presented in Figure 7 (just add an arc between step iii and step IV.)

Lines 189-90: “Altogether, these data suggest that polymerization of C9 is rate-limiting in the

assembly of complete MAC pores on bacteria.” This needs to be more clearly stated that the C9:C9 interaction is slower than the initial C8:C9 interaction. Having said this, it is impossible to reconcile this whole E.coli, ensemble data with published single molecule, time resolved data (AFM work published in ref 30) that provides more reliable evidence of the opposite kinetic observation? With this kind of ensemble data that relies on titration of protein then it isn’t really a true kinetic study.

Figure 4d: Would be complete with the Western blot of the C9 lock mutant as well.

Lines 235 - 262: There is not enough data to support the interpretation of the impaired C9 polymerisation on complement resistant E. coli being due to the improper anchoring of C5b-7. Further explanation or experiments would be needed for this section and it seems rather immature.

**Part III – Minor Issues: Editorial and Data Presentation Modifications**

Reviewer #1: The discussion is clear, although some claims made need to be tempered and some areas suggested as novel more clearly described as confirmations of the Spiller findings – for example, that C9lock binding to C5b-8 is as efficient as WT (already in the Spiller paper which goes way beyond “suggesting”). The suggestion that binding of C9 somehow increases killing capacity of C5b-8 is repeated but how this off-model effect might be explained is not discussed. Although some of the rather strong historical data on MAC killing of bacteria is mentioned (Joiner and Brown – though only a single, albeit important, paper), some highly relevant, particularly from Dankert (see above) and Taylor, data are not included and would strengthen the discussion.

Reviewer #2: A minor editorial point – there is no reference to Fig 6d in the text – I think lines 258 and 261 refer to Fig 6d (and not 6c)?

Reviewer #3: Lines 109-10: “These data confirm that the capacity of C9TMH-1 lock to form polymers is impaired, and can be reversed by reducing the cysteine bridge lock.” Figure S1c does not show the “C9TMH1 lock” forming polyC9 in reducing conditions (in SDS-PAGE) and so this statement has not been experimentally qualified. This is later covered by analysis of Fig1e so move to that section?

Lines 138-9: “Reducing C9TMH-1 lock with DTT restored its capacity to damage the IM (Fig. 2b)” Add at the end of this phrase the following or equivalent “relative to wtC9 in reducing conditions which had overall reduced activity compared to wtC9 in non-reducing conditions”

Figure 2c figure legend line 666: is this 3 or 4 “individual values (c) of three independent experiments with mean +/- SD. Also the SD bar for the far right bar looks truncated?

Line 243 and other parts of methods. Make sure that you have appropriate units after each number full stop for example 10% c5-depleted serum should have volume per volume after the 10%

Line 456 why was there no tenfold dilution?

PLOS authors have the option to publish the peer review history of their article (what does this mean?). If published, this will include your full peer review and any attached files.

Reviewer #1: No

Reviewer #2: No

Reviewer #3: No
---

## [Decision Letter · Decision Letter 1]

20 Oct 2021

Dear Dr Bardoel,

We are pleased to inform you that your manuscript 'Polymerization of C9 enhances bacterial cell envelope damage and killing by membrane attack complex pores' has been provisionally accepted for publication in PLOS Pathogens.

Best regards,

Seppo Meri

Guest Editor

PLOS Pathogens

Christoph Tang

Section Editor

PLOS Pathogens

Kasturi Haldar

Editor-in-Chief

PLOS Pathogens

orcid.org/0000-0001-5065-158X

Michael Malim

Editor-in-Chief

PLOS Pathogens

orcid.org/0000-0002-7699-2064

There were comments from reviewer 3, especially concerning the binding of "locked" C9 to C5b-8 in relation to preventing C9 polymerization, and whether this was already demonstrated first by the current authors or in the paper by Spencer et al. There is also a general comment on using an appropriately wide range of concentrations when performing lysis assays. These comments can be acknowledged, but do not - in my view - necessitate another round of revision.

Reviewer Comments (if any, and for reference):

Reviewer's Responses to Questions

**Part I - Summary**

Reviewer #1: The authors have addressed the majority of my concerns in the revision. A few minor niggles remain but are not sufficient to justify a further revision.

Reviewer #3: This is a thorough study of MAC assembly, specifically C9 binding, in the context of the bacterial surface. Overall there are some great tools used in this study and is a step closer in under the mechanism of action of the MAC on bacteria.

**Part II – Major Issues: Key Experiments Required for Acceptance**

Reviewer #1: None.

Reviewer #3: Normally I would not too critical at this stage but it appears that the authors continue to over-embellish at times. There is a major point raised by reviewer 1, Point 5.

Line 120 "Although Spicer et al. already demonstrated that locking the TMH-1 domain prevents polymerization of C9 [30], we wanted to confirm that this did not prevent binding of C9 to C5b-8." And line 305 “Our study extends on these insights, since it provides direct evidence that polymerization of C9 enhances bacterial killing by using a system in which C9 can bind to C5b-8 without polymerizing. Although Spicer et al. had already demonstrated that locking the TMH-1 domain of C9 could prevent polymerization of C9 [30], we extend on these insights by confirming that this does not affect binding of C9 to C5b-8 with direct binding experiments on E. coli.”

Technically the FACS is still only looking at the binding of the added C9 to the cells with implication that it is binding to the MAC and is not really a direct protein:protein binding experiment or a SP experiment. I would say that this overall result is only a slight enhancement on the lysis activity only based work presented in the Spicer paper where the competition assay and the structure suggest that the first C9 will bind (locked or WT).

But I also accept that this at the next stage of revision and will take guidance from the editorial team.

ANSWER 19 (also reviewer 1, point 4)- We have added the full original experiment which included a concentration range of C9wt and C9TMH-1 lock (S1b). Spicer et al. measured only at one C9 concentration in a different experimental set-up, which makes it possible that they did not pick up this small haemolytic effect of C9TMH-1 lock.

The reviewer thanks the authors for adding the full experiment. Spicer et al use a different type of assay where they observe lysis over time which has more information compared to standard assays but, as the authors point out, do not do this on a range of concentrations. So, it would be good to point out to both the authors and the editors that in future all lysis activity assays (regardless of the type) should use a suitable range of concentrations. This should become standard for high quality journals, such as PLoS Pathogens, for future publications. It would be good if the pore forming protein community ensures that this is the standard.

**Part III – Minor Issues: Editorial and Data Presentation Modifications**

Reviewer #1: None

Reviewer #3: (No Response)

PLOS authors have the option to publish the peer review history of their article (what does this mean?). If published, this will include your full peer review and any attached files.

Reviewer #1: No

Reviewer #3: No

---

## [Editor Report · Acceptance letter]

4 Nov 2021

Dear Dr Bardoel,

We are delighted to inform you that your manuscript, "Polymerization of C9 enhances bacterial cell envelope damage and killing by membrane attack complex pores," has been formally accepted for publication in PLOS Pathogens.

Best regards,

Kasturi Haldar

Editor-in-Chief

PLOS Pathogens

orcid.org/0000-0001-5065-158X

Michael Malim

Editor-in-Chief

PLOS Pathogens

orcid.org/0000-0002-7699-2064